# Genome-wide sequencing identifies a thermal-tolerance related synonymous mutation in the mussel, *Mytilisepta virgata*

Yue Tan[1,4], Chao-Yi Ma[1,4], Xiao-Xu Li[1], Guo-Dong Han[2] & Yun-Wei Dong [1,3✉]

The roles of synonymous mutations for adapting to stressful thermal environments are of fundamental biological and ecological interests but poorly understood. To study whether synonymous mutations influence thermal adaptation at specific microhabitats, a genome-wide genotype-phenotype association analysis is carried out in the black mussels *Mytilisepta virgata*. A synonymous mutation of Ubiquitin-specific Peptidase 15 (*MvUSP15*) is significantly associated with the physiological upper thermal limit. The individuals carrying GG genotype (the G-type) at the mutant locus possess significantly lower heat tolerance compared to the individuals carrying GA and AA genotypes (the A-type). When heated to sublethal temperature, the G-type exhibit higher inter-individual variations in *MvUSP15* expression, especially for the mussels on the sun-exposed microhabitats. Taken together, a synonymous mutation in *MvUSP15* can affect the gene expression profile and interact with microhabitat heterogeneity to influence thermal resistance. This integrative study sheds light on the ecological importance of adaptive synonymous mutations as an underappreciated genetic buffer against heat stress and emphasizes the importance of integrative studies at a microhabitat scale for evaluating and predicting the impacts of climate change.

---

[1] The Key Laboratory of Mariculture, Ministry of Education, Fisheries College, Ocean University of China, Qingdao 266003, PR China. [2] College of Life Science, Yantai University, Yantai 264005, China. [3] Function Laboratory for Marine Fisheries Science and Food Production Processes, Pilot National Laboratory for Marine Science and Technology, Qingdao 266235, PR China. [4] These authors contributed equally: Yue Tan, Chao-Yi Ma. ✉email: Dongyw@ouc.edu.cn

Understanding the capacity of natural populations to adapt to changing thermal environments is of key interest in global change studies. For a long time, genetic variations have been considered the most important component of adaptive evolution in species[1,2]. In light of climate change, mounting studies of thermal adaptation demonstrate certain genetic variations do facilitate the matching of species' tolerant abilities with the warming conditions in their immediate habitats[3,4]. For most species with limited dispersal abilities, a need exists to counter stress from within-site warming conditions if local extinction is to be avoided; the adaptation resulting from genetic polymorphism can play an important role as an initial buffer[5,6]. While investigating the adaptive effects of genetic factors, it is often found that these factors have interactions with environmental heterogeneity and generate joint influence in adaptive evolution[7,8]. Furthermore, the importance of fine-scale variability in both abiotic and biotic factors gets increasingly highlighted in the ecological studies on thermal adaptation[9]. However, comprehensive studies measuring the interactive influences between adaptive genetic variations and fine-scale thermal environments on adaptation to certain microhabitats remain scarce. Such studies should be developed for evaluating and predicting the resistance or vulnerability of species in response to anthropogenically driven warming conditions.

Synonymous mutations may be an underappreciated driver for adaptation to within-site thermal conditions. This class of genetic variations is often described as mutations in evolutionary dynamics but can still influence phenotypic traits[10]. As this kind of mutation leads to non-alteration in the amino acid sequence of the encoded protein, the mechanisms by which synonymous mutations confer biological effects are presumably via changes to multiple stages for the gene expression in vivo, including transcriptional process[11], mRNA folding, and decay[12,13], and protein translation[14]. Furthermore, there have been studies providing direct evidence demonstrating that synonymous mutations in certain genetic regions can have fitness effects and contribute to adaptive responses[15]. Bailey et al. reported two independently arising synonymous mutations that could induce upregulation of gene expression that had strong beneficial fitness effects and could drive adaptive evolution in a laboratory bacterial population[16]. Agashe et al. found some large-effect beneficial synonymous mutations consistently had positive impacts during laboratory evolution, showing that these mutations can mediate rapid adaptation. However, the effects of these mutations depended on the local sequence context[17].

Although the significance of adaptive synonymous mutations has been confirmed and speculated to be common in natural populations, the question remains as to whether these mutations have ecologically important effects related to thermal adaptation. When undergoing extreme thermal stress, those individuals carrying adaptive synonymous mutations associated with thermal tolerance can possess higher fitness[18]. While having potentially wide importance, the adaptive effects of synonymous mutations remain understudied. Much further work is required to determine how synonymous mutations affect the way species respond to heat stress within habitats and what the broad role of these mutations is in the context of the genetic basis of thermal adaptation.

Promising studies on how fine-scale variations in habitat conditions influence thermal adaptation under the pressure of global change have utilized species from intertidal zones[19,20], where extreme environmental heterogeneity is found over small spatial and temporal scales[21]. For example, it has been shown that in intertidal ecosystems, thermal variations among microhabitats can generally reach or even exceed the differences found over much larger geographic scales like variations of latitudes[22]. This high heterogeneity of habitat conditions at the microhabitat scale, however, poses serious challenges for the survival of intertidal ectothermic animals, which have to develop adaptive variations at all levels of biological organization, ranging from genetic structure[23], to physiological responses[24,25] and behavioral modes[26,27] to maximize their fitness at specific microhabitats. For this reason, intertidal ectotherms with their thermal-adaptive traits are considered a model system for elucidating the mechanisms that provide species with different capacities for adaptation to heat stress[28]. Among these animals, the black mussel *Mytilisepta virgata* (Wiegmann, 1837), a widely distributed species native to the Indo-west-Pacific, may serve as a model species for investigating fine-scale adaptive traits. It has been reported that when experiencing high temperatures in summer, different microhabitats present widely different levels of heat stress. Thus, while *M. virgata* can be safe in shaded microhabitats that serve as thermal refugia with benign temperatures, the individuals inhabiting the sun-exposed microhabitats live near their thermal limits and have to endure long-term sublethal heat stress in summer[24]. Adaptive genetic variations are conjectured to be critical to its survival under warming[4]. Thus, investigation of the microhabitat-specific differentiation in the genetic adaptation of *M. virgata* can provide insights into the fine-scale genetic adaptation to thermal stress.

Two hypotheses were addressed in the present study: (1) adaptive synonymous mutations are closely related to the heat tolerance of the mussel and (2) these synonymous mutations interact with fine-scale thermal heterogeneity for providing microhabitat-specific thermal adaptation. To address these hypotheses, we first obtained a fine-scale temperature dataset during field experiments to elucidate the thermal variation among microhabitats. We next employed phenotype-genotype association analysis to find adaptive mutations related to thermal tolerance and then conducted experiments to assess the impacts of the synonymous mutation on the gene expression. In support of our predictions, we found that synonymous mutation can affect the thermal tolerance of *M. virgata* by changing related gene expression levels and identified the correlation between the genetic impacts and the microhabitat heterogeneity. Our results support the conjecture that synonymous mutations are crucial for surviving heterogeneities in thermal environments and should be considered in evaluating and predicting the impacts of climate change.

## Results

**Thermal heterogeneity in different microhabitats.** There were significant differences in the operative temperatures in summer between the sun-exposed and shaded microhabitats where two data logger iButtons were deployed (Paired Mann–Whitney test, $V = 586862$, $p$ value < 0.05). To estimate the level of thermal stress within habitats, the 99th percentile of all temperatures recorded in every month ($T_{99}$) and the average daily maximum temperature (ADM) during the month were calculated. The $T_{99}$ was the monthly highest temperature that the mussels are likely to experience and was used as a measure of "acute" thermal stress in each month, while ADM was considered as a measure of "chronic" high-temperature exposure[20]. During the summer months, including July and August in 2020, both the $T_{99}$ and ADM at the sun-exposed microhabitats were higher than those at the shaded microhabitats. The value of $T_{99}$ in the sun-exposed microhabitat was 42.13 °C, while in the shaded microhabitat, the value was 34.30 °C. The value of ADM in the sun-exposed microhabitat was 35.13 °C, while in the shaded microhabitats was 31.33 °C (Fig. 1a and Supplementary Data 1).

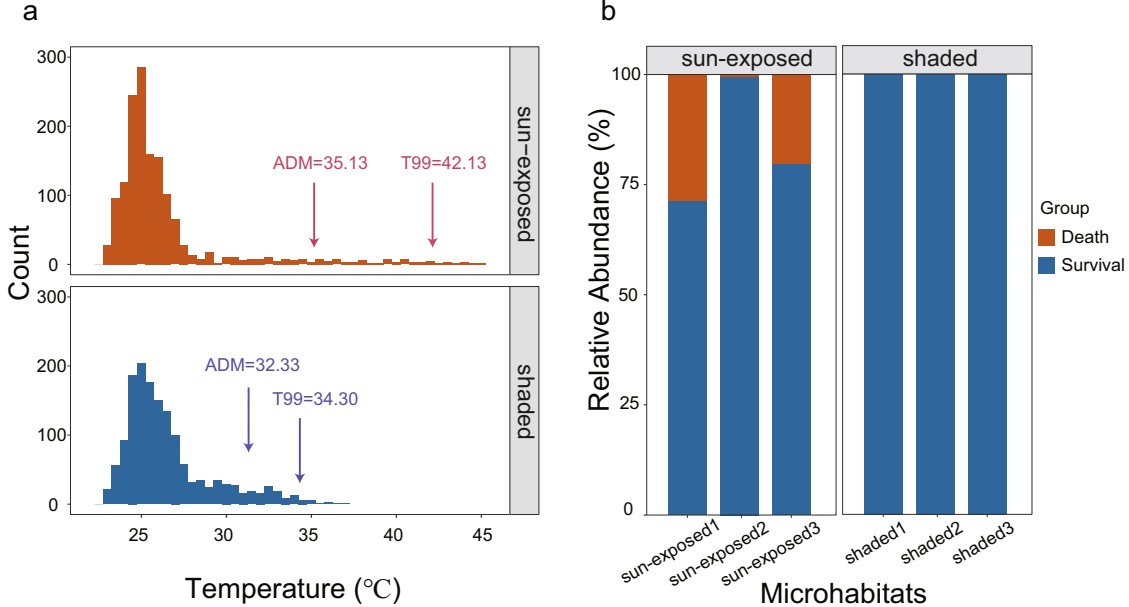

**Fig. 1 Comparison of thermal environments from the native habitats of *Mytilisepta virgata*. a** In situ operative temperature frequency diagram of sun-exposed and shaded microhabitats in July and August 2020. **b** The morality observed of *M. virgata* individuals at the sun-exposed and shaded microhabitats.

In keeping with these site-related differences in thermal stress indicators, higher mortality was observed in the sun-exposed than in shaded microhabitats (Supplementary Fig. 1) (Mann–Whitney test, $W = 9$, $p$ value = 0.032). The mortality rates in three separate sun-exposed microhabitats were 28.74, 0.56, and 20.26%, while no deaths occurred in the shaded microhabitats (Fig. 1b).

**Thermal tolerance-related synonymous mutation**. To clarify the role of genetic differences among individuals in establishing upper thermal tolerance, a genome-wide association mapping analysis was performed. A total of 24,703 SNPs was obtained and used for the genotype-phenotype association analysis, and the ABT, the sublethal thermal limit as indicated by cardiac performance, was used as a thermally-sensitive phenotypic trait.

A synonymous mutation in the seventh exon of *MvUSP15* was significantly associated with the ABT of *M. virgata* (1881663_28, genomic-control corrected $p$ value = 8.13e$^{-6}$) (Fig. 2a). We also found that there were four mutant sites within the genomic region of *MvUSP15* gene; these were, the locus 1881663_64 in the seventh exon of *MvUSP15*, the loci 1881700_56, 1881700_132, and 1881700_150 in the intron of *MvUSP15*. However, the loci 1881663_64, 1881700_56, and 1881700_132 had been pruned out by SNP filtering and thus not included in the association analysis. The remained site 1881700_150 showed no significant correlation with heat tolerance (genomic-control corrected $p$ value = 0.4578). Besides, this synonymous mutation was not located at some other sequence element that overlapped with the coding sequence of *MvUSP15*, as there were no overlapping genes with *MvUSP15* within its genomic regions. Thus, we mainly focused on the effects of this synonymous mutation on the expression of *MvUSP15* in this study.

The mutation site 1881663_28 is attributed to the transition from G to A at the locus, which leads to an alteration in serine codon usage from UCG to UCA. Depending on the alleles carried, the mussels were divided into the G-type (mussels with homozygous GG genotype) and the A-type (mussels with homozygous AA genotype and heterozygous GA genotype) (Fig. 2b). The mean ABT of the A-type (44.34 ± 0.99 °C, mean ± s.d.) was significantly higher than the G-type (42.29 ± 1.10 °C, mean ± s.d.) (Two-sample *t*-test, $t = 6.0268$, df = 20.80, $p < 0.01$) (Fig. 2c). Apart from the locus 1881663_28, three other sites showed significant correlation with heat tolerance of mussels; these were, the locus 2020699_47 at pseudo-chromosome 13 (genomic-control corrected $p$ value = 1.55e$^{-5}$), 1652060_107 at pseudo-chromosome 11 (genomic-control corrected $p$ value = 7.699e$^{-5}$), and 1796187_33 at pseudo-chromosome 12 (genomic-control corrected $p$ value = 7.278e$^{-5}$). However, these loci were located in the introns of genes whose functions could not be determined based on the genome annotation files and BLAST results. No strong linkage disequilibrium relationship was found between the synonymous mutant locus 1881663_28 and locus 1796187_33 ($D' = 0.2772$, $r^2 = 0.033$).

Dramatic seasonal fluctuation in genotype frequency existed at the locus 1881663_28. The proportions of A-type mussels were 5.00% in April, increased to 52.38% in August, and fell back to 10.53% in December (Fig. 2d). The results of two-way ANOVA showed that the sampling month and genotype could significantly affect ABT and no significant interaction between the sampling month and genotype (Supplementary Table 1). The mussels used in the association analysis were further divided into four clusters using a two-step cluster analysis (Table 1), in which the genotype (G- or A-type) and the sampling month (April, August, and December) were set as categorical variables while ABTs of mussels were set as continuous variables. There were significant differences in ABT among these four clusters (one-way ANOVA analysis, $SS = 68.01$, df = 3, $F = 43.74$, $p$ value < 0.05). The multiple comparison results among four clusters were listed in Supplementary Table 2.

**Characterization of *MvUSP15*. *MvUSP15* was a single-copy gene located on chromosome 12 of *M. virgata* (Supplementary Fig. 2). This gene with 21 exons spanned ~39 kb of genomic DNA, in which the full-length CDS region was 2772-bp long and encoded a protein of 923 amino acids with an estimated molecular mass of 106.3 kDa and an isoelectric point of 5.29 (Supplementary Fig. 3).

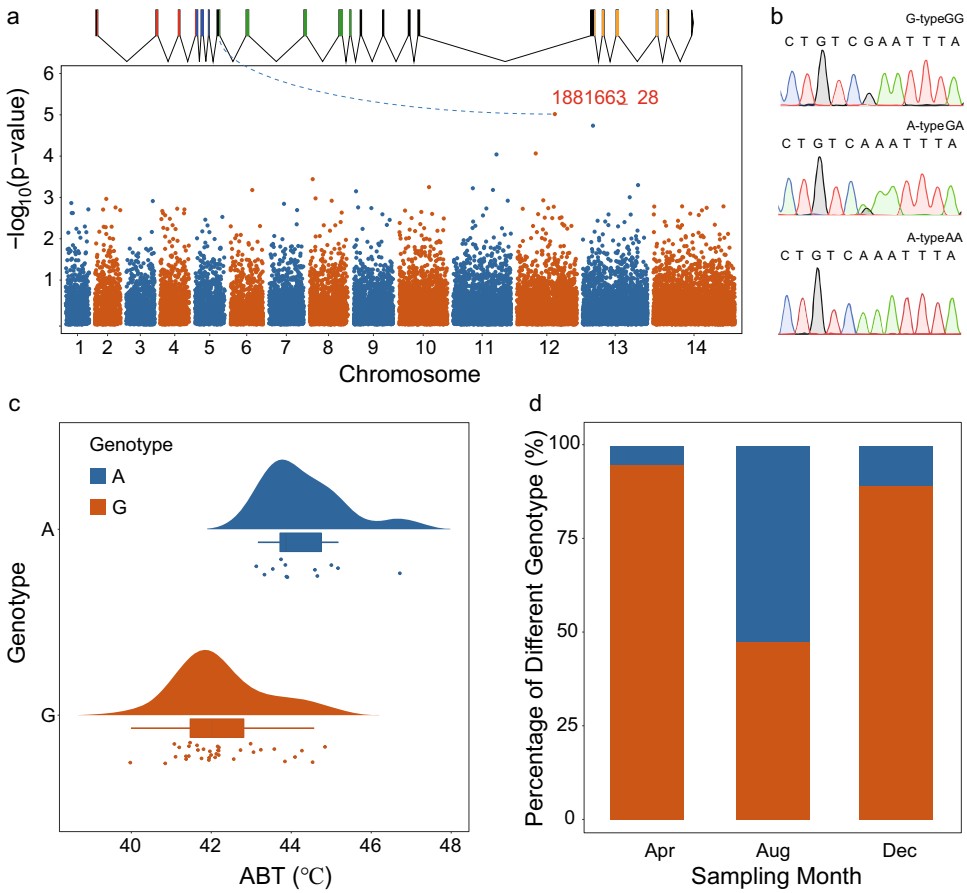

**Fig. 2 Thermal tolerance-related synonymous mutation and its frequency in *M. virgata* populations. a** Genome-wide association mapping analysis between *M. virgata* genotype and thermal tolerance of cardiac performance (ABT). The Manhattan plot was applied to show the statistical significance of SNPs in the genome-wide association study. Each point represents one SNP. The genomic-control corrected *P* value for each SNP's association was shown on the y-axis. One locus 1881663_28, located at the seventh exon of the *USP15* gene, had a corrected *p* value below 1e-5. **b** The sequence diagrams of G-type (genotype GG at locus 1881663_28) and A-type (genotype GA / AA) individuals. **c** The probability density and scatter plot by *M. virgata* heat tolerance. Each point represents the ABT of an individual with a G-type (red) or A-type (blue) genotype. **d** The seasonal variations in the proportions of G-type (red) and A-type (blue) individuals in April (Apr), August (Aug), and December (Dec).

**Table 1 Four clusters of *M. virgata* individuals in association analysis using two-step cluster analysis.**

| Cluster | 1 | 2 | 3 | 4 |
|---|---|---|---|---|
| Individual numbers | 12 | 8 | 9 | 19 |
| Sampling month | | December | August | April |
| ABT (°C)* | 44.34 ± 0.99[a] | 41.63 ± 0.81[b] | 43.88 ± 0.67[a] | 41.82 ± 0.46[b] |
| Genotype | A-type | G-type | G-type | G-type |

The genotype (G- or A-type) and sampling month are set as categorical variables, while ABT (mean ± s.d.) is set as continuous variables. Cluster 1 contains ten individuals, one individual, and one individual in August, April, and December, respectively.
*Different superscript letters in a row represent significant differences using one-way ANOVA.

Results of the analysis on codon usage bias demonstrated that UCA was a more frequently used serine codon (RSCU value = 1.3) than UCG (RSCU value = 0.3) in *MvUSP15* with respect to the synonymous codon usage differentiation (Fig. 3a).

The analysis of the *MvUSP15* protein was conducted using the deduced amino sequence from the verified *MvUSP15* CDS. The analysis of domain architectures showed the *MvUSP15* protein consisted of a DUSP (domain present in ubiquitin-specific proteases) which was one of the specific subdomains of deubiquitinating enzymes (DUB), a ubiquitin-like (UBL) domain which could be found in several ubiquitin carboxyl-terminal hydrolases, and a long carboxy-terminal catalytic domain[29] (Supplementary Table 3).

A phylogenetic tree was constructed with amino acids sequences of the *MvUSP15* encoding protein and USP15 protein in other organisms (Fig. 3b and Supplementary Data 2), using data downloaded from the Refseq databases (https://www.ncbi.nlm.nih.gov/refseq/). These proteins from species across the taxa in Bilateria shared several conserved sequence motifs (Fig. 3b) and domains (Supplementary Fig. 4), indicating a high degree of conservation. Furthermore, the USP15 sequences of Lophotrochozoan species, including the mussel *Mytilus edulis*, the scallop *Mizuhopecten yessoensis*, the land snail *Candidula unifasciata*, the sea slug *Plakobranchus ocellatus*, the freshwater snail *Pomacea canaliculate*, *Capitella teleta* (Annelida), and *Lingula anatine* (Brachiopoda), were clustered together and different from the

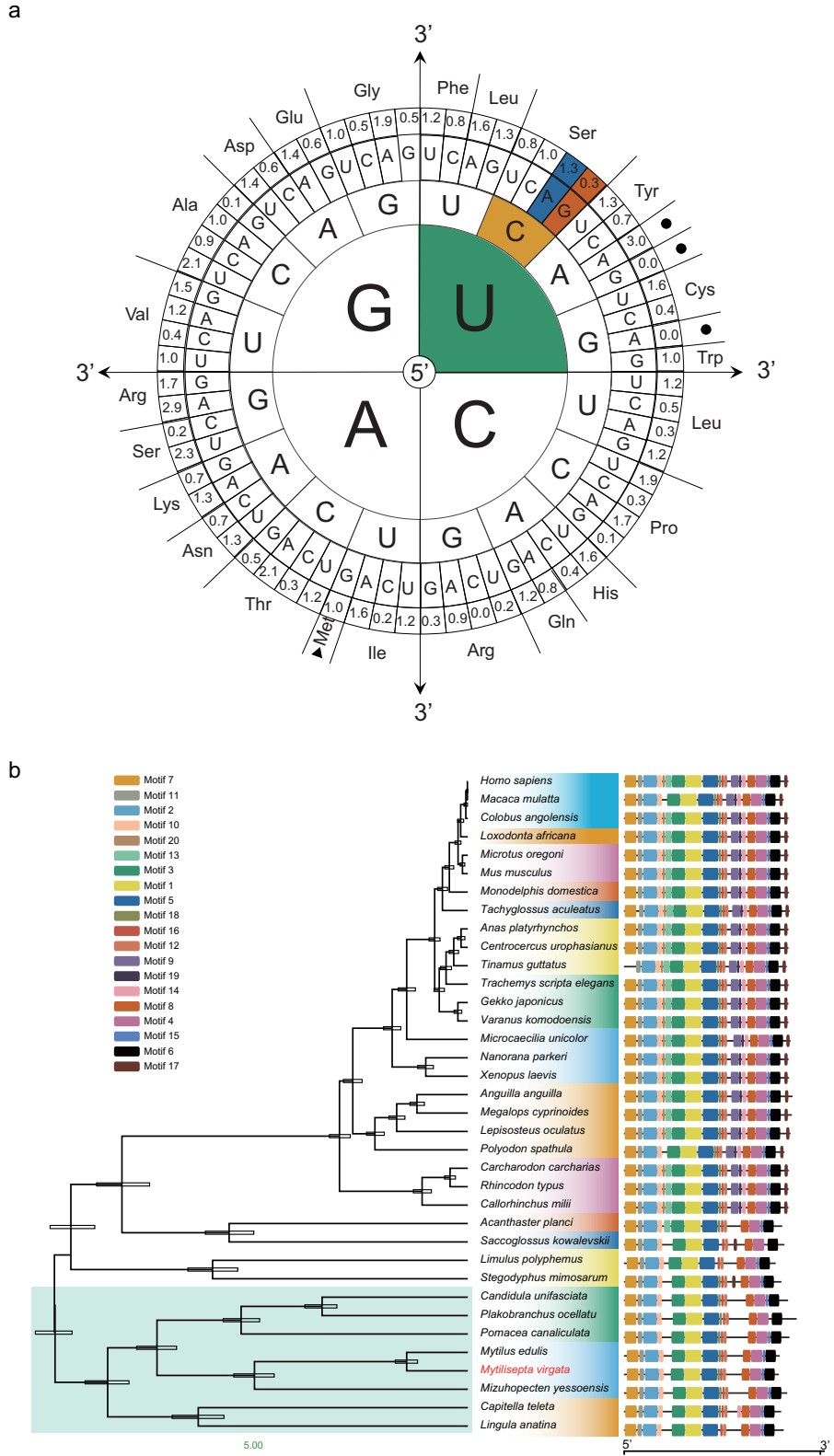

**Fig. 3 Identification and characterization of *MvUSP15*. a** The codon usage in the *MvUSP15*. RSCU-values of the codon used were listed. The values of the UCA codon and UCG were 1.3 and 0.3, respectively. **b** Phylogenetic tree based on the deduced *USP15* amino acid sequence. The phylogenetic tree was constructed using Bayesian evolutionary analysis. Numerical values on the branches represent the scaled differentiation time. Those branches with light blue backgrounds represent species from Lophotrochozoan. Different motif architectures were shown in different colors.

other two major clades of bilaterian animals, namely Arthropoda and Chordata, indicating that USP15 arose earlier than previously reported[30].

**Influences of synonymous mutation on the mRNA stability and levels of *MvUSP15*.** The mRNA with G at the synonymous mutant site had a lower $\Delta G_{fold}$ (higher stability) compared with the mRNA with A at that site at most temperatures (Supplementary Fig. 5). With the temperature rising, the effects of the synonymous mutation on $\Delta G_{fold}$ differed between the G and A containing orthologs are significant (Paired Mann–Whitney test, $V = 1370.5$, $p$ value < 0.05), indicating the single synonymous mutation having influences on mRNA stability.

The expression levels of *MvUSP15* in the heat-treated group (53.96 ± 38.05, mean ± s.d.) significantly increased compared with the control group (42.77 ± 20.39, mean ± s.d.) at 22 °C (Two-sample $t$-test, $t = -1.7193$, df = 49.381, $p$ value < 0.05) (Fig. 4 and Supplementary Data 3, 4). In the heat-treated group, the *MvUSP15* expression level in G-type mussels (56.55 ± 42.50, mean ± s.d.) was higher than in A-type mussels (49.07 ± 27.91, mean ± s.d.) but not significantly different (Two-sample $t$-test, $t = -0.9157$, df = 69.539, $p$ value = 0.1815). The absolute deviation of expression levels of *MvUSP15* in G-type mussels (32.69 ± 26.74, mean ± s.d.) was significantly higher than in A-type mussels (23.74 ± 13.89, mean ± s.d.) (Two-sample $t$-test, $t = -1.9086$, df = 72.99, $p$ value < 0.05), indicating a higher inter-individual variation in *MvUSP15* expression levels of the G-type experiencing heat stress by reaching sublethal temperature (Supplementary Fig. 6). The individuals in the heat-treated group

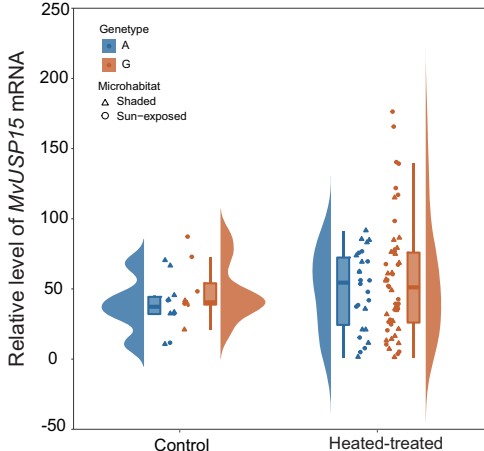

**Fig. 4 Expression profile of *MvUSP15* mRNA of the mussels from the heat-treated group and control group.** Different colors represent different genotypes. Each triangle and circle represent the relative level of *MvUSP15* mRNA of one individual sampled from the shaded microhabitats and sun-exposed microhabitats, respectively.

could be divided into five clusters (Table 2) using the two-step cluster analysis, in which the genotype (G- or A-type) and microhabitat (sun-exposed or shaded microhabitats) were set as categorical variables, and the mRNA levels and absolute values of deviations were set as continuous variables. There were significant differences in *MvUSP15* mRNA expression levels among these five clusters (one-way ANOVA analysis, $SS = 54949$, df = 4, $F = 18.43$, $p$ value < 0.001). The expression levels of cluster 4, containing seven G-type individuals from the sun-exposed microhabitat, were significantly higher than those of the other clusters (Supplementary Table 4). These results indicated that when experiencing heat stress, part of G-type mussels from the sun-exposed microhabitats could have higher upregulated expression levels of *MvUSP15*.

**Discussion**
A single genetic mutation has the power of changing phenotypic traits by influencing the expression profile of related genes[18]. In the present study, a synonymous mutation at *MvUSP15* is found to be associated with an increased thermal resistance of *M. virgata* (as indicated by using ABT as a criterion of sublethal stress), indicating this genetic variation is an important hereditary trait contributing to the high heat resistance of the mussels. Our results further suggest that the adaptive synonymous mutations combined with the thermal heterogeneity in the natural environments can be conducive to survival at high temperatures in summer. Therefore, for evaluating and predicting the effects of high temperatures on population dynamics in the face of climate change, fine-scale genetic and physiological adaptation, including adaptive synonymous mutations occurring at certain gene regions, should be given full consideration.

The adaptation of species to changing environments can benefit from allelic polymorphism within populations[2]. Whereas most emphasis on allelic variants has focused on proteins with adaptively different amino acid sequences, we found *M. virgata* individuals with different genotypes at the synonymous mutation locus 1881663_28 had divergent capabilities of thermal tolerance. The relatively rare A-type mussels carrying the allele A have higher sublethal temperatures than the G-types, and this result is consistent with previous studies revealing that genetic variations at some loci can attribute to the changes in thermal-tolerant phenotypes[31]. Here, the greater thermal tolerance of A-type individuals led to a decrease in allele G frequency of the population in August. However, the allele G frequency rose again in December and thus showed remarkable fluctuation related to ambient temperature. Our previous study pointed out the occurrence of analogous seasonal changes of genotype frequency at some thermal-tolerant associated loci and recovery of the relative heat-sensitive genotype in the cold season, resulting from balancing selection[4]. Although *M. virgata* has a sessile lifestyle in the adult stage and the genotypic component within a population can't be randomized by the migration of adult mussels, intertidal invertebrates are usually r-strategists with high

**Table 2 Five clusters of *M. virgata* individuals in the heat-treated group using a two-step cluster analysis.**

| Cluster | 1 | 2 | 3 | 4 | 5 |
|---|---|---|---|---|---|
| Individuals number | 12 | 24 | 18 | 7 | 14 |
| Microhabitat | Shaded | Shaded | Sun-exposed | Sun-exposed | Sun-exposed |
| Genotype | A-type | G-type | G-type | G-type | A-type |
| Expression level* | 53.08 ± 32.98[b] | 48.03 ± 30.83[b] | 36.79 ± 19.86[b] | 136.61 ± 27.38[a] | 45.63 ± 23.45[b] |
| Absolute deviation* | 30.06 ± 10.93[ab] | 26.52 ± 17.10[b] | 22.50 ± 16.49[b] | 80.06 ± 27.38[a] | 18.32 ± 14.19[b] |

The genotype (G- or A-type) and microhabitat (sun-exposed or shaded microhabitats) were set as categorical variables, while *MvUSP15* expression levels (mean ± s.d.) and absolute deviations of expression levels (mean ± s.d.) were set as continuous variables.
*Different superscript letters in a row represent significant differences using one-way ANOVA.

reproductive rates and larval dispersal capabilities[32]. Thermal-tolerant heterozygotes (GA in the present case) going through thermal stress during summer would produce offspring with all three genotypes, and the loss of homozygotes GG could be compensated by reintroduction during reproduction and recruitment in the breeding season from September to December[33]. These newly produced pelagic larvae with different genotypes would randomly recruit in different microhabitats, in some of which the loss of thermal-resistant mussels in summer would have opened-up settlement space. Such a reproductive and recruitment process might lead to a balanced genetic composition for the mussel population within-site. Future work involving large-scale genotyping of the *M. virgata* population to determine if the allele frequencies differ between adults and juveniles would be helpful to test the hypothesis that balancing selection is the compensation mechanism for seasonal fluctuation of gene frequency within populations.

Two closely related questions also need to be explored further to better understand the evolutionary histories and functional importance of the A and G alleles. First, it would be interesting to find out which allele is the "mutant", and which is the ancestral state. This information could also provide a basis for predicting future patterns in the prevalence of the A and G alleles. Thus, a second question needs deeper analysis. This question concerns whether the allele A will become the dominant allele within the population if the selection pressure for heat tolerance is a long-term trend. Based on currently available data, the reality seems to be the opposite. Thus, due to the difference between the genotype frequencies of two alleles and the finding that allele G is dominant within a population at most seasons, we propose that allele G is the wild type while allele A is the mutant type. The rise of this mutant allele within the population may result from natural selection due to thermal stress, as suggested by our finding that mussels with allele A might have a selective advantage when the population experience extremely high temperatures in summer. However, in addition to heat stress, other selectively important factors may affect genotype compositions, for example, cold stress in winter and competition and predation risks that *M. virgata* faces in its complex natural environment[34,35]. Influences of these other factors might counteract the beneficial heat tolerance effects of this adaptive synonymous mutation and might be the reason for impeding the process of allele G being gradually replaced by allele A. Nevertheless, the positive effects of this synonymous mutation on mussel populations when extremely high temperatures occur still remain an important element in selection for the genetic composition of the population. Notably, individuals possessing allelic variants that confer improved heat tolerance might serve as "seeds" that carry a population through extreme thermal events[25] and allow heat resistance and resilience at the population level.

The variations in expression levels of *MvUSP15* found in mussels with different genotypes may provide clues to the mechanism of adaptive variation at this locus. The *MvUSP15* expression profile of G-type mussels at 42 °C has much more inter-individual variation than A-type mussels; compared to A-type mussels, some G-type individuals had dramatically upregulated *MvUSP15* expression after the sublethal heat treatment. This high expression of *MvUSP15* is conjectured to impede the protective mitophagy pathway in vivo against oxidative damage induced by extremely high temperatures. This decrease in protection against one consequence of heat stress, disruption of mitochondrial homeostasis, might be part of the cause of weakened heat resistance in G-type mussels.

The family of USP15 proteins is highly conserved evolutionarily and participates in analogous cellular processes in divergent taxa. One of the established cellular physiological functions of *USP15* is serving as a powerful inhibitor of the PINK1/Parkin-mediated mitochondrial apoptosis pathway by antagonizing ubiquitination[32], while mitophagy is activated to maintain cellular homeostasis under abiotic or biotic stress[36]. Previous studies involving the thermal physiology of molluscs demonstrated that elevated levels of reactive oxygen species (ROS) in vivo often accompany the increase of temperature[37], particularly at a temperature around the ABT[38,39]. To alleviate the cellular damage caused by heat-induced oxidative stress, mitophagy pathways that remove damaged mitochondria are relevant to the thermal-tolerant responses in molluscs[40]. In this context, *MvUSP15* is conjectured to be a critical regulatory enzyme in the adaptive physiological responses of *M. virgata*. Thus, the abnormal upregulated expression of *MvUSP15* under high-temperature conditions will have adverse effects on the normal mitochondrial apoptosis pathway, whose function is important under heat stress. Thus, *M. virgata* individuals carrying the A allele at the synonymous mutant locus 1881663_28 may benefit from a relatively more consistent *MvUSP15* expression profile within the group than mussels carrying the G allele. Apart from acting as an inhibitor of mitophagy, USP15 is a widely expressed deubiquitylase that has been involved in diverse cellular processes regulating thermal tolerance. For example, the USP gene family, including USP15, has been reported to play important roles in regulating the process of chromatin condensation, which is one hallmark of apoptosis potentially induced by thermal stress[41,42]. More studies should be carried out to further investigate the specific mechanisms whereby this conserved USP15 gene family helps to establish thermal tolerance.

The synonymous mutation examined in this study, wherein the preferred codon UCA was at times replaced by the rare and non-optimal codon UCG[43], could have led to a number of influences on gene expression. For example, codon usage can have significant effects on translation because synonymous codons vary in translation efficiencies; these effects can influence important aspects of the translation process, for example, amino sequence elongation speed and co-translational protein folding[44–48].

Different synonymous codon usage can also affect gene expression levels in a translation-independent manner[11,49]. A previous study has pointed out that the biological impacts of a single synonymous mutation on mRNA levels are pleiotropic: the usage of optimal codon enhances the levels at most cases; however, in some instances, the usage of non-optimal codon resulted in higher mRNA levels[13]. The latter phenomenon was also found in the present study: although the relatively non-optimal codon UCG occurred in the CDS of *MvUSP15* of G-type mussels, this genotype had stronger upregulation of mRNA levels than A-type mussels under conditions of heat stress. A potential explanation for this result might be that the effects of synonymous codon usage on mRNA concentration could be complex and involve factors related to the post-transcriptional process[50]. For instance, due to the alteration of synonymous codon usage, the secondary structure and stability of mRNA can be affected[51], further can lead to changes in mRNA levels[50,52]. In the present study, the synonymous mutation in *MvUSP15* was shown to cause differences in $\Delta G_{fold}$ between the A- and G-type mRNAs. The mRNA containing the UCG codon had a higher absolute value of $\Delta G_{fold}$, an index for mRNA structural stability[51,53,54], which should be more stable than the sequence using the UCA codon. According to the studies of Victor et al., who investigated the influence of the codon usage pattern and mRNA structural stability on mRNA levels, higher stability of mRNA secondary structure resulting from different codon usage, often led to higher mRNA abundance[55]. Thus, the usage of the non-optimal codon UCG, which led to higher mRNA stability, may further cause the upregulated *MvUSP15* expression level in some G-type mussels

experiencing temperature increases. However, a large gap still exists in a more specific understanding of the mechanisms explaining how synonymous mutation influences gene expression; a great deal of work combining the methods investigating the differences between nuclear and cytosolic levels of RNA to better determine the mechanisms of synonymous mutation on gene expression[56,57]; and the approaches of reverse genetics such as an F2 hybrid screen with quantitative trait locus (QTL) mapping, should be useful for further investigating the role of synonymous mutations in the thermal adaptation.

The microhabitat thermal heterogeneity also affected the expression profile of *MvUSP15*. We found that some G-type mussels from the sun-exposed microhabitats exhibited abnormal upregulation of *MvUSP15* after the indoor heat shock at 42 °C; these individuals were clustered together using the Two-step cluster analysis. Microhabitat-specific variability in the gene expression profile of the Antarctic intertidal limpet *Nacella concinna* that had experienced common garden acclimation subsequent to the collection was also observed[58], indicating that the influence of recent habitat conditions on gene expression profile could not be erased easily. Thus, we conjecture that the inter-individual variations in the *MvUSP15* expression profile were not only associated with genetic effects, i.e., the adaptive synonymous mutation discussed above, but also reflected thermal heterogeneity at a microhabitat scale.

Compared with the mussels inhabiting the shaded microhabitats with benign thermal environments, the individuals in the sun-exposed habitats frequently face much more severe thermal conditions. Previous studies showed that the mussels inhabiting the sun-exposed microhabitats would experience long exposures to sublethal or lethal heat stress on the subtropical shore in summer[4,24]. The cumulative adverse effects from the high temperatures in the sun-exposed microhabitats may impair the normal *MvUSP15* expression profile in multiple ways, like epigenetic factors, which have been considered to be important in the gene expression flexibility associated with environmental heterogeneity[58,59]. Compared with the occurrence of highly variable *MvUSP15* upregulation in G-type mussels on the sun-exposed shore, extreme variation in expression was found neither in G-type mussels from the shaded microhabitats nor in A-type mussels in all microhabitats; these data indicate that an interaction between genetic effects of synonymous mutation at locus 1881663_28 and microhabitat-associated variability in environmental conditions existed.

Overall, standing genetic variations and heterogeneous abiotic characteristics among habitats can work together to interactively influence the adaptive potential of species to changing environments[60]. In the present study, a synonymous mutation of *MvUSP15* associated with inter-individual variations in thermal tolerance showed a genotype-specific and microhabitat-specific gene expression profile, implying that both genetic polymorphism and availability of benign refugia are critical for surviving harsh environments. Compared with nonsynonymous mutations, the occurrence of synonymous mutations has a much higher frequency[61]. Although the ecological importance of synonymous mutations on the adaptation of field populations has remained underappreciated, the present study, for the first time, shows that this kind of synonymous mutation can be related to thermal resistance. The combination of both adaptive synonymous mutation and microhabitat heterogeneity is considered as arming the mussels with advantageous traits against thermal stress. This work lays particular emphasis on the ecological importance of synonymous mutations and provides a potentially important new type of groundwork for future studies in climate change biology.

## Materials and methods

**Microhabitat-scale temperature and mortality statistics survey of *M. virgata* in the field**. From July to Augustus 2020, we continuously recorded the operative temperature within native habitats of *M. virgata* using biomimetic thermal loggers[62]. An iButton data logger (DS1922L, Maxim Integrated) was inserted into the shell cavity of a mussel (shell length, 4–5 cm) whose soft tissues had been removed. Then, the Robomussel was completed by filling the shell cavity with Flame Retardant Compound (Scotchcast 2130, 3 M, MN, USA) to waterproof the electronics. All Robomussels used for measuring were also calibrated using a digital thermometer (Fluke 54II, Fluke, WA, USA). The Robomussels were then deployed at a field site at the Dongshan Swire Marine Station (D-SMART), China (23.65°N, 117.49°E), in which two types of microhabitats coexisted at rocky shores. One representing a refuge completely sheltered by rocks (shaded microhabitats), and the other occurred in full sunlight (sun-exposed microhabitats). In each microhabitat where the *M. virgata* had a high abundance, two Robomussels were deployed between 1.80 and 2.50 m above the chart datum (CD, the level from which depths displayed on a nautical chart are measured). The temperature was recorded at an interval of 1 h and an accuracy of 0.5 °C. The 99th percentile of all temperatures recorded ($T_{99}$) was calculated as the highest temperature occurring within habitats in each month, and the average daily maximum temperature in each month (ADM) was calculated as the average of all daily peaks in each month[20].

The mortality of the mussels was calculated via quadrat methods in microhabitats where the Robomussels were deployed in July 2020. A total of 18 quadrats (25 cm × 25 cm) were deployed (3 quadrates × 2 types of microhabitats × 3 microhabitats), and the number of *M. virgate* in each quadrate was counted. A mussel was regarded as alive if the shell kept closed or closed immediately when it was touched, whereas it was regarded as dead if the mussel was empty or had no response when it was touched. Then the mortality was calculated and the difference in mortality between the sun-exposed and the shaded microhabitat was compared using statistical tests.

**Heart rate measurement and sequencing library construction**. From February to December 2016 and every 2 months, *M. virgata* individuals were randomly sampled on the rocky shore in Dongshan Island, Fujian Province (117°29' E, 23°39'N). Then the mussels were transported back to the laboratory within three hours and immersed in fresh seawater with a temperature of ~20 °C and a salinity of ~33 psu to simulate the seawater in situ. Before performing the heart rate measurements, a 3-day common garden acclimation was conducted, during which seawater was aerated continuously and exchanged daily and the mussels were fed daily with concentrated *Chlorella*. For the non-invasive heart rate measurement experiments[63], the designated temperatures increased at a rate of 6 °C h$^{-1}$ in the air from acclimation temperature (~20 °C) until the heart rate fell to zero. For detecting the heartbeat of mussels, an infrared sensor fixed (with Super Glue, Loctite, USA) to the mussel shell at a position above the heart (next to the mid-dorsal posterior hinge area) was used. Then we recorded the amplified and filtered variations in the light-dependent current produced by the heartbeat using an infrared signal amplifier (AMP03, Newshift, Portugal) and PowerLab AD converter (8/30, ADInstruments, Australia), and viewed and analyzed the heartbeats using LabChart v7 (ADInstruments, Australia). The Arrhenius Breakpoint temperature (ABT) for cardiac performance is the temperature after which the HR decreases sharply with constant heating, was determined using a regression analysis method that generates the best fit line on either side of a putative break point for the relationship of ln-transformed heart rate (beats per minute) against absolute temperature and calculated using segmented package in R version 4.1.0[64,65]. ABT temperature was an indicator for determining when anaerobic metabolism is activated by sublethal physiological stress and has been used as a common index for evaluating sublethal thermal tolerance in molluscs[24–26].

For identifying single nucleotide polymorphisms (SNPs) in a total of 64 extracted DNA samples (21 collected in April, 23 collected in August, and 20 collected in December), genome-wide double digest restriction site-associated DNA sequencing (ddRADseq) was performed. We prepared the sequencing libraries based on a protocol adapted from the McDaniel Laboratory at the University of Florida (https://mcdaniellab.biology.ufl.edu/data/), in which the EcoRI and MspI restriction enzymes (New England Biolabs, MA, USA) were used. Libraries were then sequenced as 150 bp paired-end reads using Illumina NovaSeq at Berry Genomics Corporation (Beijing, China), and deposited in BioProject (PRJNA517974) and BioSample (SAMN10849586–SAMN10849649) databases at NCBI (National Center for Biotechnology Information).

**Genome-wide association mapping analysis**. The ddRADseq data described above and our *M. virgata* genome (BioProject: PRJNA910323) were used for identified SNPs in the 64 samples at the genome-wide level. The reads were first cleaned using fastp[66], aligned to reference with the BWA-MEM algorithm[67], and sorted using SAMtools[68]. The BAM outputs were then processed via gstacks script of package Stacks[69] to create loci and identify SNPs with unpaired reads discarded (--rm-unpaired-reads). Candidate loci were then filtered and converted to VCF file using the *populations* script of the package Stacks. The following parameters, including a maximum number of missing samples per locus of 20% (-r 0.8), a minimum number of populations of three (-p 3), a minor allele frequency of 0.05

(-min_maf 0.05) by populations script, and a maximum per-SNP missing percentage of 10% (--geno 0.1), and an exact Hardy–Weinberg disequilibrium $p$ value less than $1e^{-6}$ (--hwe 1e-6) by PLINK version 1.90 program[70] were used for determining whether a locus could be retained. Then those retained SNPs were further pruned by performing linkage disequilibrium (LD) based SNP pruning utilizing the PLINK option "--indep-pairwise 50 5 0.2". Filtered loci were functionally annotated with ANNOVA[71]. Finally, the genotype-phenotype association analysis using 48 out of 64 sample's ABT values as thermal-tolerant phenotypic traits was performed with PLINK. To further reduce the possibility of false positives, the $p$ values of those associated SNPs were adjusted using the genomic-control method as described by ref. [72]. The function of the gene at which the mutant sites were located was first confirmed by referring to the annotation file of the $M.\ virgata$ genome, and then the exact function of the gene was further classified based on BLAST results, molecular phylogeny, and inspection of conserved domains.

**Cloning and Sequencing of the $M.\ virgata$ USP15 gene.** Total RNA was extracted from the adductor muscle of one mussel using TRIzol reagent following the manufacturer's protocol (Invitrogen, China). The RNA purity was determined by measuring the A260/A280 ratio on a Nanodrop spectrophotometer, while RNA integrity was visually assessed by using 1% (w/v) agarose gel electrophoresis. The cDNA was synthesized from isolated RNA by using a PrimeScript™ II 1st Strand cDNA Synthesis Kit (TaKaRa, China), following the manufacturer's protocol, and used for PCR-amplification with the primer pair (uch-CDS) based on the annotated $M.\ virgata$ USP15 gene ($MvUSP15$ gene) from the $M.\ virgata$ genome (Supplementary Table 5). PCR reactions using LA Taq enzyme (TaKaRa, China) were performed on the thermal cycler (Biometra) with an initial denaturation step of 94 °C for 3 min, followed by 39 amplification cycles consisting of denaturation at 94 °C for 30 s, annealing at 51 °C for 30 s, elongation at 72 °C for 2 min and 30 s, and the reactions were terminated with an elongation step of 72 °C for 10 min, followed by cooling at 4 °C. The amplified products were first purified using an AxyPrep™ DNA Gel Extraction Kit (Corning, China), followed by an independent ligation process into the pMD19-T Vector system (TaKaRa, China). Then the recombinant plasmids were transformed into $Escherichia\ coli$ Stellar Competent Cells (Clontech). The cells were cultured on LB plates containing ampicillin. The positive transformants were sent to Sangon Biotech (https://www.sangon.com/) for Sanger sequencing.

**Sequence analysis of $MvUSP15$.** Verified $MvUSP15$ coding domain sequences (CDS) were used for subsequent analysis and visualization of the nucleic acid sequence. R package $RIdeogram$[73] was used for mapping genome-wide information, including gene density and the distribution of the gene family encoding deubiquitinating enzymes on the idiogram. The visualization of the $MvUSP15$ gene structure was performed using the online tool Exon-Intron Graphic Maker (http://www.wormweb.org/exonintron). DnaSP software was used for the codon usage bias analysis and calculation of relative synonymous codon usage (RSCU) values based on the verified CDS sequence of $MvUSP15$[74].

The deduced amino acid from the verified $MvUSP15$ CDS sequence was used for subsequent analysis of the $MvUSP15$ protein. The ExPASy ProtParam tool (https://web.expasy.org/protparam/) was used for molecular mass and isoelectric point calculations. The analysis of protein domain architectures was conducted using the SMART (Simple Modular Architecture Research Tool) website (https://smart.embl.de)[75]. Database homology searching was performed using the BLAST program at the NCBI website (https://blast.ncbi.nlm.nih.gov/Blast.cgi). The corresponding sequences were obtained from GenBank and then aligned using MEGA X software (https://www.megasoftware.net/)[76]. Evolutionary analysis and Bayesian phylogenetic tree construction were performed using MCMC methods, which were conducted with BEAST v2.6.6 software[77]. Motif structure analysis for every protein sequence was conducted on the MEME website (https://meme-suite.org/meme/doc/meme.html)[78].

We compared the stability of two orthologous mRNAs of $MvUSP15$ with the base difference (G or A) at synonymous mutation 1881663_28. The free energy change occurring during the formation of the ensemble of secondary structures (ΔGfold) was used as an index for estimating the stability of mRNA structure[51,55]. We first performed the free energy minimization structure prediction for two orthologous mRNAs from 22 °C (295.15 K) to 52 °C (325.15 K) using the Fold program[79] with default parameters in the RNA structure package. The ΔGfold for two orthologous mRNAs was calculated by using the efn2 algorithm[80]. Calculations were performed with default parameters at different temperatures (22° to 52 °C) along the predicted secondary structure determined using the Fold program.

**Sample collection, acclimation, and heat treatment of $M.\ virgata$ individuals.** Mussels were collected in both the shaded and sun-exposed habitats along the shoreline in D-SMART on July 31st, 2021. Specimens collected in the two microhabitats were acclimated for two months, a period likely to be sufficient to achieve the fully acclimated state[81,82]. During laboratory acclimation, the mussels from shaded and sun-exposed microhabitats were held separately in two plastic containers and immersed in fresh seawater at the water temperature of ~20 °C and salinity of 33 psu. Seawater was aerated continuously and exchanged daily. The mussels were fed on concentrated $Chlorella$ every 2 days. After acclimation, the mussels from each microhabitat were randomly allocated into a heat-treated group or a control group. In the heat-treated group, mussels were heated in the air using a water bath (TXF 200; Grant, UK) from the acclimation temperature (22 °C) to 42 °C, the sublethal temperature for $M.\ virgata$[24], at a rate of 6 °C h$^{-1}$ for simulating the stressful thermal environment during emersion in summer, while the control group was kept at 22 °C in air. After the heating process, both the heat-treated and control group were dissected and stored in liquid nitrogen.

**Genotyping for $M.\ virgata$ individuals.** Total DNA was extracted from the foot muscle of the mussels in both the heat-treated group and control group with the CTAB method following the protocol based on $Littorina\ saxatilis$[83]. The DNA purity was determined by measuring the A260/A280 ratio with a Nanodrop spectrophotometer (Thermo Scientific, China), and the DNA integrity was visually assessed by 1% (w/v) agarose gel electrophoresis.

PCR-amplification reactions using the primer pair (GT-454) designed based on $MvUSP15$ (Supplementary Table 5), were conducted with the rTaq enzyme (TaKaRa, China) using a thermal cycler (Bio-Rad, China) with an initial denaturation step of 94 °C for 3 min, followed by 35 amplification cycles consisting of denaturation at 94 °C for 30 s, annealing at 40 °C for 30 s, and elongation at 72 °C for 30 s. The reactions were terminated with an elongation step of 72 °C for 3 min, followed by cooling at 4 °C. The amplified 454-bp products containing the synonymous mutation locus were sent to Sangon Biotech (https://www.sangon.com/) for Sanger sequencing. A total of 179 mussels were genotyped successfully. Sanger sequencing files were viewed in SnapGene software (https://www.snapgene.com/) and the genotypes of those 179 mussels were recorded in Supplementary Data 3.

**mRNA Expression of $MvUSP15$ by quantitative real-time reverse transcription PCR.** The quantitative real-time PCR (qRT-PCR) experiments were conducted to examine the level of $MvUSP15$ mRNA concentration, of which the whole process met MIQE standards[84]. Total RNA was extracted from the adductor muscle of mussels in both the heated-treated group and the control group using TRIzol reagent following the manufacturer's protocol (Invitrogen, China), and cDNA was synthesized using a PrimeScript RT reagent Kit with gDNA Eraser (Takara, China). The qRT-PCR analysis was conducted to quantify the $MvUSP15$ transcript levels on a CFX96 Real-Time PCR Detection System (Bio-Rad). Specific primers (Qpcr1) for $MvUSP15$ were designed based on the identified sequence. The elongation factor 1α (EF-1α), was chosen as the reference gene for internal standardization (Supplementary Table 5)[85]. These two pairs of primer sequences were validated by melting curve analysis to ensure there were no nonspecific product amplification and primer-dimer formation. The PCR efficiencies for EF-1α and $MvUSP15$ were calculated by standard curve analysis and were 108.2 and 105.2%, respectively. The qPCR reaction was performed using 2× SYBR Premix Ex Taq (TaKaRa, China) following a protocol for a CFX96 Real-Time PCR Detection System (Bio-Rad): 95 °C for 30 s, followed by 40 cycles of 95 °C for 5 s and 60 °C for 30 s, then 95 °C for 10 s; and the melt curve analysis was then conducted with 0.5 °C increments from 65 to 95 °C for 5 s as per the standard procedure. A no-template control (NTC) was also performed to preclude any extraneous DNA or RNA template from providing a false signal. The NTC thus served as a control for extraneous nucleic acid contamination and as an important control for primer-dimer formation. Each sample was tested in triplicate and the relative expression level of $MvUSP15$ was calculated using the $2^{-\Delta\Delta Ct}$ method[86]. The calculation of relative $MvUSP15$ expression levels was conducted using Bio-Rad CFX Manager software (https://www.bio-rad.com/).

**Statistics and reproducibility.** All calculations and comparative analyses were performed using R version 4.1.0[65] and Two-Step cluster analyses were conducted with SPSS 25 (SPSS, Chicago, IL, USA). A total of 93 mussels were used to compare $MvUSP15$ expression differences under different microhabitats (sun-exposed or shaded) and heat treatment (heat-treated or control) conditions. For all figures and tables, a $p$ value less than 0.05 was used as the criterion of significant difference in all tests.

**Reporting summary.** Further information on research design is available in the Nature Research Reporting Summary linked to this article.

## Data availability

The genome dataset was deposited in GenBank under the accession number PRJNA910323. In addition, we uploaded annotation files of the $M.\ virgata$ genome, including gene locations (.gff3), protein models (.pep), and CDS (.cds) sequences to FigShare online (https://figshare.com/articles/dataset/Genome-wide_sequencing_identifies_a_thermal-tolerance_related_synonymous_mutation_in_the_mussel_Mytilisepta_virgata/21699641).

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

## Acknowledgements

This study is supported by the National Natural Science Foundation of China (42025604) and the Fundamental Research Funds for the Central Universities. We thank Prof. George Somero of Stanford University for his constructive suggestions and discussions. We thank YongXu Sun of Xiamen University for his assistance in sample collection and also thank Qisi Cai and Shengyao Sun at the Dongshan Swire Marine Station of Xiamen University for their logistical support.

## Author contributions

Y.-W.D. designed research; G.-D.H. provide partial data; C.-Y.M. and X.-X.L. performed research; Y.-W.D., Y.T., and C.-Y.M. analyzed data; Y.-W.D., Y.T., and C.-Y.M. wrote the paper.

## Competing interests

The authors declare no competing interests.
