## [Peer Review File · Communications Biology]

Reviewers' comments:

Reviewer #1 (Remarks to the Author):

The manuscript entitled "Adaptive synonymous mutation interacts with microhabitat heterogeneity to influence thermal resistance" provides an interesting example of a synonymous mutation that appears to contribute to an adaptive phenotype ("appears" as I do not know the statistical methods well enough to judge how well the causal relationship is established). In a genome-wide genotype to phenotype association study a single synonymous mutation in the gene *USP15* in the mussel *M. virgata* was picked out as being correlated with survival during heat stress. The allele frequency at the mutant locus varied across the season as could be expected for mutations that affect thermal tolerance.

I have a few concerns, questions, and comments:

1. I am a bacterial geneticist and not familiar with genome wide association mapping in general and mussel population biology in particular. Given the relatively small number of individuals sequenced (64) and the large number of SNPs (24703) it sounds unlikely to me that no other loci were picked up randomly in the association analysis? How likely is it to get a false positive hit with a significant p-value from such a dataset? In Figure 2 there are several SNPs that "stand out" from the rest. Was "1881663_28" the only one that was significant, or did it just have the lowest p-value and you chose to ignore the other ones? – I guess my question stem from my own ignorance about the methods, but perhaps the authors need to clarify this?

2. The data show a correlation between which allele is present in the *USP15* locus and survival during heat stress. The case for a causal relationship is strengthened by the apparent seasonal variation in the relative allele frequencies, but in order to judge if this apparent seasonal variation is real I think we need more information:

How likely is it that this variation was due to sampling error?

How many samples were taken at each time-point?

Does the data allow you to exclude the hypothesis that the *USP15* mutation is merely genetically linked to another mutation which causes the phenotype?

What is the mechanism of the rapid changes in allele frequencies over the season? I could imagine that if the G type dies off in large numbers during the hot season, that the A-type increases in relative frequency. But what is the mechanism of recovery of the G type in the cold season? How mobile/stationary are these mussels? Are they expected to migrate fast enough so that the population is "randomized" during the cold season, or are they so stationary that dead ones are replaced by young mussels or larvae (I assume the larvae are more mobile than the adult mussels)? Would the allele frequencies differ between young and old individuals within the same microhabitat (could be a factor depending on how the dead mussels are replaced)? Could this be tested in the current dataset? Would the selection pressure for heat tolerance lead to a long-term trend that the A-type increases over longer time-scales, or is there some other selection pressure (in the cooler microhabitats) that counteracts the beneficial effect of the A-type?

If any of these questions makes sense, maybe they could be addressed in the discussion?

3. Perhaps just to satisfy my curiosity: Another thing that is not entirely clear is which is the "mutant" and which is the ancestral state? The G or the A? Is the mutation recent or has it been part of the population for a long time? Is it possible to draw any conclusions from other populations of *M. virgata*? Or from closely related species?

4. That a synonymous mutation can cause a phenotype (and that the phenotype may be adaptive) is not far-fetched and not very new. However, as the sequence you show in Figure S3 is from a cDNA I think we are missing some information in order to judge if the mutation is really "just synonymous".

An important piece of information that is missing is exactly where in the sequence of the MvUSP15 gene the mutation is located? The text says the mutation is a UCG (serine) → UCA (serine) synonymous mutation in the seventh exon, but it could be interesting to know more precisely. Perhaps you could mark the position of the mutation and exon boundaries in Figure S3? Are there any other functional sites in the DNA or mRNA that could be of interest? How is chromatin condensation regulated (and can it be influenced by a single SNP? ignore this if it doesn't make sense with the what's known about the different levels of gene regulation in eukaryotes).

5. Figure S3: The amino acid sequence is not aligned to the coding sequence. Perhaps the font was converted from a monospace font to a variable width font when generating the figure from a text file?

6. Figure 3 (b and c): I do not see what these analyses add to the current manuscript? What does a picture of the structure of USP15 or a phylogenetic tree of USP15 protein from different species have to do with the genotype to phenotype association of the synonymous mutation? I do not see that these contribute anything to this manuscript, and the authors should consider removing them (or provide arguments for why they are necessary in case I missed it in the text). Other figures/tables that I do not find any motive for including are Figure S4, Table S2, Table S6. Or are these used in order to establish that the locus "1881663_28" is really the *M. virgata* USP15 ortholog? In that case this should be clarified in the text.

7. While the data show a correlation between the allele at USP15 and USP15 gene expression variability in heat stressed mussels, the mechanism for this is not clear. The authors mention that "Codon usage bias between UCA and UCG contributes to genotypical differentiation in the transcription of MvUSP15". What is this supposed to mean? How would a single synonymous mutation affect transcription? The discussion further mentions codon usage bias and transcription in the same sentence, without any mention of the most obvious level at which codon usage bias is expected to have an effect: translation. Are the UCA and UCG codons expected to be translated at different efficiencies?

8. The manuscript could need some improvement of the English language. Most sentences are ok, but here are some examples of sentences that needs to be improved. Note that I have not looked very carefully and have probably missed several other similar language problems:

Line 389 - "One of the convincing cellular physiological functions of USP15 has been shown as a powerful inhibitor against PINK1/Parkin-mediated mitochondrial apoptosis pathway via ubiquitination antagonizing" (I guess it is meant to say "by antagonizing ubiquitination"?)

Line 413 - "Another study focusing on synonymous mutations of IGF-1 in mammals suggested that the occasional using rare codons inducing slightly upregulation in the transcriptional level might due to the usage of a rare codon relieved the pressure of commonly used codons to a certain extent" (there are several problems with the structure and grammar of the sentence that makes it difficult to figure out exactly what it is meant to say).

Line 407 - "Gerdol et al. (2015) found species belonging to Mytilidae had codon usage bias towards UCA codon rather than UCG codon and our characterization analysis of the *MvUSP15* cDNA sequence was consistent with this opinion" (the word "opinion" makes it look like there is a matter of debate with regards to the codon bias that Gerdol et al reported. I guess it is just a poor choice of words, and that "observation" would be a better choice?)

Reviewer #2 (Remarks to the Author):

MAJOR COMMENTS:

I greatly appreciate the efforts and work of the authors and do believe they have very interesting findings toward a genetic basis of heat tolerance in this *Mytilus* species. However, I find two logical flaws in their interpretation of the results. Without these flaws addressed by additional experimental work, my suggestion is the rejection of the manuscript in its current form.

1. First, the authors show no evidence that the synonymous polymorphism in MvUSP15 are the mechanistic basis of the of the differential thermal tolerance observed. I find it at least as probably that the genetic basis of the phenotype is another site that is in tight linkage with the identified locus.

2. Second, the authors posit that codon usage impacts MvUSP15 expression and examine expression at the transcriptional level. Codon bias typically affects translation, but I did follow up on the author's discussion of Wang et al. (2019; <https://doi.org/10.1016/j.omtn.2021.08.007>), in which mRNA stability appears to be impacted by codon usage.

Additional experiments to map the mechanistic locus underlying the phenotype could include an F2 hybrid screen with QTL mapping using a WGS approach to identify candidate loci at higher resolution than the current RAD-based mapping, ideally followed up by reverse genetics to test the candidates.

It appears that forward genetics are not yet published for *Mytilus* species, but in the absence of genetic manipulation, there are additional experiments which could be performed, perhaps examining mRNA stability by blocking transcription using Actinomycin-D and assessing mRNA levels.

MINOR COMMENTS:

How was the MvUSP identified from the candidate loci? Are there additional candidate loci? How large is the window based upon knowledge of linkage in this/related taxa?

Regarding the qPCR experiments, do the experiments meet MIQE standards?

Is EF-1 α stable and suitable for use as a housekeeping/internal control gene based on empirical data from this species?

The explanation of T99 and ADM would improve the clarity of the manuscript.

Manuscript will benefit from judicious copyediting overall.

Point-by-point responses to comments of reviewers “*Adaptive synonymous mutation interacts with microhabitat heterogeneity to influence thermal resistance*”:

Reviewer #1:

*The manuscript entitled “Adaptive synonymous mutation interacts with microhabitat heterogeneity to influence thermal resistance” provides an interesting example of a synonymous mutation that appears to contribute to an adaptive phenotype (“appears” as I do not know the statistical methods well enough to judge how well the causal relationship is established). In a genome-wide genotype to phenotype association study a single synonymous mutation in the gene USP15 in the mussel *M. virgata* was picked out as being correlated with survival during heat stress. The allele frequency at the mutant locus varied across the season as could be expected for mutations that affect thermal tolerance.*

I have a few concerns, questions, and comments:

Response:

Thank you for your comments. We have carefully considered your suggestions and tried our best to improve the manuscript. Please find our responses and modifications as follows.

Q1. I am a bacterial geneticist and not familiar with genome wide association mapping in general and mussel population biology in particular. Given the relatively small number of individuals sequenced (64) and the large number of SNPs (24703) it sounds unlikely to me that no other loci were picked up randomly in the association analysis? How likely is it to get a false positive hit with a significant p-value from such a dataset? In Figure 2 there are several SNPs that “stand out” from the rest. Was “1881663_28” the only one that was significant, or did it just have the lowest p-value and you chose to ignore the other ones? – I guess my question stem from my own ignorance about the methods, but perhaps the authors need to clarify this?

Response to Q1:

All these comments and suggestions are valuable and important for improving our manuscript.

(1) A false positive hit with a significant P value may exist. For avoiding this problem, we have taken this concern into account during data analysis and employed methods for eliminating the existence of such false positives. Firstly, after being randomly collected from the natural environment, the mussels experienced common garden acclimation to eliminate influences other than genetic factors before performing experiments to measure heart rate. Secondly, the genomic association analysis methods described by Devlin *et al.* have been used to further control the generation of false positives¹. Finally, the p-value corrected by the Bonferroni correction also showed that the loci of interest in the study (Bonferroni corrected p-value = 0.08129) had the strongest significance

We have now stated these points more clearly in the text of the Methods and Materials section in the revised manuscript:

Page 6-7, line 145-152, “From February to December 2016, *M. virgata* specimens were randomly collected every 2 months on rocky shore in Dongshan Island, Fujian Province (117°29' E, 23°39'N). After collection, mussels were transported back to the laboratory within 3h, placed in a plastic basket, and immersed in 20 l fresh seawater with temperature of ~20°C and salinity of ~33 psu to simulate the in situ annual average water temperature for a 3-day acclimation period before performing the heart rate measurements. Seawater was aerated continuously and exchanged daily. Mussels were fed daily with concentrated *Chlorella*.”

Page 8, line 197-199, “For further reducing the possibility of false positive, the p-values of those associated SNPs were adjusted using the genomic control method as described by Devlin *et al.*”

References

1. Devlin, B. & Roeder, K. “Genomic control for association studies.” *Biometrics*. **55**, 997-1004 (1999).

(2) As to why the 1881663_28 site was selected, the following criteria were used: (i) this site has the highest p-value adjusted by correction methods as described above. (ii) this site is located on the seventh exon of the Chr12.1841 gene, so the mutation of this site is very likely to directly affect the expression of the gene. (iii) Chr12.1841 has been annotated with a relatively clear function, namely ubiquitin carboxyl-terminal hydrolase 15, which is involved in pathways such as mitophagy, a process that is considered to be associated with heat tolerance.

The reasons why this study did not focus on other sites with significant p-values ($p\text{-value} < 1e^{-4}$) (2020699_47, 1796187_33, 1652060_107) are as follows: (i) These sites are located within introns of genes, and although many studies have pointed out that changes in introns can affect gene expression²⁻⁴, mainly through changes of RNA splicing, it is difficult for us to verify such effects in a non-model species like *M. virgata*. (ii) More importantly, the annotation results for the genes at which the mutant sites are located are not clear; without these identifications, we lack adequate information about how these genes might contribute to the heat tolerance of mussels. However, we are considering these genes with ambiguous functions as an area for future study and we modified the results section in our manuscript by adding more information on these sites as follows:

Page 14, line 364-373, “Apart from the locus 1881663_28, three other sites showed significant correlation with heat tolerance of mussels; these were, the locus 2020699_47 at pseudo-chromosome 13 (genomic-control corrected p-value = $1.55e^{-5}$), 1652060_107 at pseudo-chromosome 11 (genomic-control corrected p-value = $7.699e^{-5}$), and 1796187_33 at pseudo-

chromosome 12 (genomic-control corrected p-value = $7.278e^{-5}$). However, these loci were located in the introns of genes whose functions could not be determined based on the genome annotation files and BLAST results. No strong linkage disequilibrium relationship was found between the synonymous mutant locus 1881663_28 and locus 1796187_33 ($D' = 0.2772$, $r^2 = 0.033$).”

References

2. Vaz-Drago, R., Custódio, N. & Carmo-Fonseca, M. Deep intronic mutations and human disease. *Hum Genet.* **136**, 1093–1111 (2017).
3. Jung, H., Lee, K.S. & Choi, J.K. Comprehensive characterisation of intronic mis-splicing mutations in human cancers. *Oncogene.* **40**, 1347–1361 (2021).
4. Tanaka S, Hayashi T, Terada C, Hori Y, Han KS, Ahn HS, Bourre F, Tani Y. Glanzmann's thrombasthenia due to a point mutation within intron 10 results in aberrant splicing of the beta3 gene. *J Thromb Haemost.* **1(11)**:2427-33 (2003).

Q2. The data show a correlation between which allele is present in the USP15 locus and survival during heat stress. The case for a causal relationship is strengthened by the apparent seasonal variation in the relative allele frequencies, but in order to judge if this apparent seasonal variation is real I think we need more information:

How likely is it that this variation was due to sampling error?

How many samples were taken at each time-point?

Does the data allow you to exclude the hypothesis that the USP15 mutation is merely genetically linked to another mutation which causes the phenotype?

What is the mechanism of the rapid changes in allele frequencies over the season? I could imagine that if the G type dies off in large numbers during the hot season, that the A-type increases in relative frequency. But what is the mechanism of recovery of the G type in the cold season?

How mobile/stationary are these mussels? Are they expected to migrate fast enough so that the population is “randomized” during the cold season, or are they so stationary that dead ones are replaced by young mussels or larvae (I assume the larvae are more mobile than the adult mussels)?

Would the allele frequencies differ between young and old individuals within the same microhabitat (could be a factor depending on how the dead mussels are replaced)? Could this be tested in the current dataset?

Would the selection pressure for heat tolerance lead to a long-term trend that the A-type increases over longer time-scales, or is there some other selection pressure (in the cooler microhabitats) that counteracts the beneficial effect of the A-type?

If any of these questions makes sense, maybe they could be addressed in the discussion?

Response to Q2:

Thanks for your valuable comments and suggestions on these diverse issues. We have responded to them one by one and made several changes in our manuscript per your recommendations.

Response to *How likely is it that this variation was due to sampling error?:*

To reduce the influence of sampling error, we strictly obeyed the principles and methods of *Simple Random Sampling*, in which each member of the population has an exactly equal chance of being selected. In our work, we randomly collected mussels in shaded and sun-exposed habitats along a ~500 m shoreline during every sampling process.

Page 6, line 145-146, “From February to December 2016, *M. virgata* specimens were randomly collected every 2 months on rocky shore in Dongshan Island, Fujian Province (117°29’ E, 23°39’N).”

Response to *How many samples were taken at each time-point?:*

We collected 64 samples, 21 individuals in April, 23 individuals in August, and 20 individuals in December, for measuring heart rate and ddRAD library construction.

Page 7, line 170-173, “For identifying single nucleotide polymorphisms (SNPs) in a total of 64 extracted DNA samples (21 collected in April, 23 collected in August, and 20 collected in December), double digest restriction site-associated DNA sequencing (ddRADseq) was performed.”

Response to *Does the data allow you to exclude the hypothesis that the USP15 mutation is merely genetically linked to another mutation which causes the phenotype?:*

For addressing this issue, we did the following things. First, we have done a linkage disequilibrium-based SNP pruning before the association analysis. The loci that were linked to each other within a 50 Kb window have been filtered out (in the window size of 50Kb, 5Kb is the step size, and r^2 is greater than 0.2). Second, we have determined that the synonymous mutant locus 1881663_28 has no strong linkage disequilibrium relationship with the locus of 1796187_33 at the same pseudo-chromosome 12 ($D' = 0.2772$, and $r^2 = 0.033$). Third, the linkage relationship among pseudo-chromosome was not taken into consideration in this study. Although there are studies investigating the long-distance linkage disequilibrium between chromosome^{5,6}, the mussel genome we used is a draft genome, which contains 14 pseudo-chromosomes of which the numbers are arranged in ascending order of chromosome size, and thus do not have true biological meanings. For this reason, the linkages among these three loci were not considered in this study.

Page 8, line 193-195, “The retained SNPs were further pruned by performing Linkage Disequilibrium (LD) based SNP pruning utilizing the PLINK option “--indep-pairwise 50 5 0.2””.

Page14, line 371-373, “No strong linkage disequilibrium relationship was found between the synonymous mutant locus 1881663_28 and locus 1796187_33 ($D' = 0.2772$, $r^2 = 0.033$).”

References

5. Sved JA. The covariance of heterozygosity as a measure of linkage disequilibrium between

- blocks of linked and unlinked sites in Hapmap. *Genet. Res.* **93**, 285–290 (2011).
6. Koch, E., Ristroph, M., and Kirkpatrick, M. Long Range Linkage Disequilibrium across the Human Genome. *PLoS One.* **8**, e80754 (2013).

Response to other questions in Q2:

(1) Response to: *What is the mechanism of the rapid changes in allele frequencies over the season? and what is the mechanism of recovery of the G type in the cold season?*

Based on our previous research ⁷, rapid changes in other allele frequencies over the seasons also existed. In our opinion, this phenomenon resulted from balancing selection: thus, although the high temperatures during the summer could eliminate some mussels with the GG genotype, which confers the potential of heat sensitivity, when the temperature conditions return to benign in the breeding season from September to December (the cold season) ⁸, the mussels with heterozygous genotype GA could produce offspring with all three genotypes. Therefore, the loss of mussels with homozygous genotype GG during the summer could be compensated by reintroduction during subsequent reproduction and recruitment during cooler fall/winter periods. We elaborate on this possibility in our modified manuscript as follows:

Page 18, line 465-475, “Our previous study pointed out the occurrence of analogous seasonal changes of genotype frequency at some thermal-tolerant associated loci and recovery of the relative heat-sensitive genotype in the cold season, resulting from balancing selection. Although *M. virgata* has a sessile lifestyle in the adult stage and the genotypic component within a population can’t be randomized by the migration of adult mussels, intertidal invertebrates are usually r-strategists with high reproductive rates and larval dispersal capabilities. Thermal-tolerant heterozygotes (GA in the present case) going through thermal stress during summer would produce offspring with all three genotypes, and the loss of homozygotes GG could be compensated by reintroduction during reproduction and recruitment in the breeding season from September to December”

(2) Response to: *How mobile/stationary are these mussels? Are they expected to migrate fast enough so that the population is “randomized” during the cold season, or are they so stationary that dead ones are replaced by young mussels or larvae (I assume the larvae are more mobile than the adult mussels)?*

The mussel *M. virgata* has two life history stages, a pelagic larval stage and a sessile adult stage, so the adult mussels can’t move freely. And as you commented, the pelagic larvae with different genotypes would randomly recruit in microhabitats, both sun-exposed and shaded, which may lead to an initially uniform genetic composition for newly settled mussels within a site.

Page 18, line 468-472, “Although *M. virgata* has a sessile lifestyle in the adult stage and the genotypic component within a population can’t be randomized by the migration of adult mussels,

intertidal invertebrates are usually r-strategists with high reproductive rates and larval dispersal capabilities.”

(3) Response to: *Would the allele frequencies differ between young and old individuals within the same microhabitat (could be a factor depending on how the dead mussels are replaced)? Could this be tested in the current dataset?*

In our collections, we selected adult mussels on the criterion of uniform size, and we don't possess age information on these specimens. Because several factors can affect size at a given age (food availability, for example, and possibly genetic differences in capacity for growth), specimens of the same size no doubt vary in age. Our focus on post-settlement section focused on temperature and microhabitat and did not include age *per se*. As we discussed above, the loss of G-type mussels during the summer will be balanced by reintroduction of G-allele containing animals during reproduction and recruitment, and pelagic larvae with different genotypes would randomly occupy vacant habitats. We added this comment into the discussion section of our manuscript:

Page 18, line 480-483, “Future work involving large-scale genotyping of the *M. virgata* population to determine if the allele frequencies differ between adults and juveniles would be helpful to test the hypothesis that balancing selection is the compensation mechanism for seasonal fluctuation of gene frequency within populations.”

References

7. Han, G., Wang, W. & Dong, Y. Effects of balancing selection and microhabitat temperature variations on heat tolerance of the intertidal black mussel *Septifer virgatus*. *Integr. Zool.* **15**, 416-427 (2020).
8. Morton, B. The biology and functional morphology of *Septifer bilocularis* and *Mytilisepta virgata* (*Bivalvia: Mytiloidea*) from corals and the exposed rocky shores, respectively, of Hong Kong. *Reg. Stud. Mar. Sci.* **25**, 100454 (2019).

(4) Response to: *Would the selection pressure for heat tolerance lead to a long-term trend that the A-type increases over longer time-scales, or is there some other selection pressure (in the cooler microhabitats) that counteracts the beneficial effect of the A-type?*

As you commented, the mussels have to face many other kinds of living pressures from nature besides thermal stress. We agree with you that some factors may suppress the possibility of A-type becoming the dominant position with the population. In the original manuscript, because we mainly focused on how the single synonymous mutation can influence heat tolerance of mussels, we neglected to consider the effects of other factors on the frequency of the A-type allele. To address this omission, we have added new text in the revised paper:

Page 18-19, line 488-500, “Thus, a second question needs deeper analysis. This question

concerns whether the allele A will become the dominant allele within the population if the selection pressure for heat tolerance is a long-term trend. Based on currently available data, the reality seems to be the opposite. Thus, due to the difference between the genotype frequencies of two alleles and the finding that allele G is dominant within population at most seasons, we propose that allele G is the wild type while the allele A is the mutant type. The rise of this mutant allele within population may result from natural selection due to thermal stress, as suggested by our finding that mussels with allele A might have a selective advantages when population experience extremely high temperatures in summer. However in addition to heat stress, other selectively important factors may affect genotype compositions, for example, cold stress in winter and competition and predation risks that *M. virgata* faces in its complex natural environment”

Q3. Perhaps just to satisfy my curiosity: Another thing that is not entirely clear is which is the “mutant” and which is the ancestral state? The G or the A? Is the mutation recent or has it been part of the population for a long time? Is it possible to draw any conclusions from other populations of M. virgata? Or from closely related species?

Response to Q3: Thanks for your insightful inquiry on this issue, which also interests but puzzles us. The wild type, as we understand it, is generally defined on the basis of the frequency of the alleles in the natural population. Any time an allele’s frequency is the majority (more than 50%), an allele is considered the wild type; conversely, the less frequent allele(s) are regarded as the mutant type(s). However, if the mutant type confers an advantage on survival and becomes dominant within the population, it becomes the wild type. In this study, we found that in 2016, apart from the August sample in which the percentage of A allele is more than 50% (52.38%), the percentage of A is relatively low, i.e., in April and December is 5% and 10.53%, respectively. Besides, according to our statistical analysis based on mussels collected in July 2021 for genotyping and validating the expression pattern of *MvUSP15*, the percentage of the A allele (~ 13.14%) is also less than G allele. However, these data are ambiguous, and lack statistical significance and we decided not to put them in our manuscript. Thus, by conventional criteria, the G allele should be regarded as the wild type and the A allele as the mutant type. However, as we have not performed research on the genetic structure of other *M. virgata* populations and our current database also has no information about the history of this allele. Thus, we cannot confirm which allele is the ancestral state and present the allele’s history within populations. We also followed your suggestion to compare the sequence of *MvUSP15* with the *USP15* gene sequences of other bivalves. Due to the reason that there have been few studies on the characterization of *USP15* genes in molluscs, all that we can do is to find the homologous sequence from annotated molluscs genomes. Multiple sequence alignments showed that the region containing the mutation sites were not conservative in the sequence of base

composition. For these reasons, we still can't draw well-supported conclusions about the history of changes at this site during evolution in closely related species. Still, your comments point out the direction for our future research; we will pay more attention to evolutionary history of the adaptive mutation. We have also added this aspect in the discussion section in our manuscript:

Page 18-19, line 484-497, “Two closely related questions also need to be explored further to better understand the evolutionary histories and functional importance of the A and G alleles. First, it would be interesting to find out which allele is the “mutant”, and which is the ancestral state. This information could also provide a basis for predicting the future patterns in prevalence of the A and G alleles... Thus, due to the difference between the genotype frequencies of two alleles and the finding that allele G is dominant within population at most seasons, we propose that allele G is the wild type while the allele A is the mutant type. The rise of this mutant allele within population may result from natural selection due to thermal stress, as suggested by our finding that mussels with allele A might have a selective advantages when population experience extremely high temperatures in summer.”

Q4. That a synonymous mutation can cause a phenotype (and that the phenotype may be adaptive) is not far-fetched and not very new. However, as the sequence you show in Figure S3 is from a cDNA I think we are missing some information in order to judge if the mutation is really “just synonymous”. An important piece of information that is missing is exactly where in the sequence of the MvUSP15 gene the mutation is located? The text says the mutation is a UCG (serine) → UCA (serine) synonymous mutation in the seventh exon, but it could be interesting to know more precisely. Perhaps you could mark the position of the mutation and exon boundaries in Figure S3? Are there any other functional sites in the DNA or mRNA that could be of interest? How is chromatin condensation regulated (and can it be influenced by a single SNP? ignore this if it doesn't make sense with the what's known about the different levels of gene regulation in eukaryotes).

Response to comments about FigS3: Thanks for your appropriate suggestions for improvement of the article's scope and readability. We have modified the Figure S3 as you advised. The synonymous mutant site and exon boundaries have been marked.

Response to *Are there any other functional sites in the DNA or mRNA that could be of interest?*

We checked the results of our genome-wide association mapping analysis, which showed that the other four base site changes within the CDS region of *MvUSP15* do exist in our original VCF file from *populations* program in STACKS, that is 1881663_64, 1881700_56, 1881700_132, 1881700_150. However, the loci 1881663_64, 1881700_56, 1881700_132 were pruned out during the LD-based SNP data filtering process. Among these three mutant sites, the loci 1881663_64 is the site most likely being another site could be interesting for its possible linkage relationship with

1881663_28. However, we also using the SNP data before LD-based SNP data filtering (only using a minimum number of populations of three, a maximum number of missing samples per locus of 20%, and a minor allele frequency of 0.05, a maximum per-SNP missing percentage of 10% and exact Hardy-Weinberg disequilibrium p-value less than $1e^{-6}$) to perform the association analysis again and found its sites to have no significant link with heat tolerance (genomic-control corrected p-value = 0.006328) because the accepted genome-wide statistical significance threshold of p-value around $1e^{-6}$ was used in this work. The remaining locus 1881700_150 after LD-based SNP data filtering shows no significant correlation with heat tolerance (genomic-control corrected p-value = 0.4578), and the same condition occurs in its linked site: 1881700_56, 1881700_132. As these results thus show, these three mutant sites have no significant correlations with the heat tolerance of mussels.

Page, 14, line 351-357, “We also found that there were four mutant sites within the genomic region of *MvUSP15* gene; these were, the locus 1881663_64 in the seventh exon of *MvUSP15*, the loci 1881700_56, 1881700_132 and 1881700_150 in the intron of *MvUSP15*. However, the loci 1881663_64, 1881700_56, 1881700_132 had been pruned out by SNP filtering and thus not included in association analysis. The remained site 1881700_150 showed no significant correlation with heat tolerance (genomic-control corrected p-value = 0.4578).”

Response to *How is chromatin condensation regulated (and can it be influenced by a single SNP? ignore this if it doesn't make sense with the what's known about the different levels of gene regulation in eukaryotes).*

Thanks for raising the issue about how a SNP can influence a biological process like chromatin condensation *in vivo*, which could be a potentially important aspect of SNPs. We have consulted relevant literature about regulation of chromatin condensation to gain insights on the question you have raised. Chromatin condensation is a protective mechanism taking part in mitosis¹⁰, as well as one of the hallmarks of apoptosis¹¹. Thus, chromatin condensation involves compacting centimetre-long DNA molecules within the confines of micrometre-sized nuclei into stable chromosomes in order to withstand the forces generated during segregation. There are quite a few ways that cells that regulate this biological process. For example, in diverse organisms ranging from prokaryotes to eukaryotes, the chromosomal condensin complex, which is a major molecular effector of chromosome condensation and segregation, is assembled for maintaining the stability of condensation process¹⁰. Thus, in our opinion, just a single base change is unlikely to have the power to cause changes in chromatin conformation during condensation. However, if the mutant took place on the key genes encoding the regulators of chromatin condensation, it could potentially influence the whole process. For example, Ura *et al.* (2001) found an amino acid substitution at the nuclear export signal region in the C-terminal site of mammalian STE20-like kinase 1 (MST1), which is a

serine threonine kinase that is capable of inducing apoptotic morphological changes such as chromatin condensation upon overexpression ¹¹.

Furthermore, other literature suggests a potential relationship between the synonymous mutant site and chromatin condensation. We found that deubiquitinating enzymes ¹², including USP15 could play a part in regulating this pathway ¹³. And, as thermal stress could induce apoptosis, which acts as a positive protective mechanism, there may exist a pathway such that, when extreme high temperature occurs, *MvUSP15* can influence apoptosis by participating in chromatin condensation. We have added a short section in the discussion section of the manuscript to comment on this issue:

Page, 20, line 536-541, “Apart from acting as an inhibitor of mitophagy, USP15 is a widely expressed deubiquitylase that has been involved in diverse cellular processes regulating thermal tolerance. For example, the USP gene family, including USP15, has been reported to play important roles in regulating the process of chromatin condensation, which is one hallmark of apoptosis potentially induced by thermal stress”

References

9. Thadani, R., Uhlmann, F., and Heeger, S. Condensin, Chromatin Crossbarring and Chromosome Condensation. *Curr. Biol.* **22**, R1012-R1021 (2012).
10. Lu, Z., Zhang, C., and Zhai, Z. Nucleoplasmin regulates chromatin condensation during apoptosis. *Proc. Natl Acad. Sci. USA.* **102**, 2778-2783 (2005).
11. Ura, S., Masuyama, N., Graves, J.D., and Gotoh, Y. Caspase cleavage of MST1 promotes nuclear translocation and chromatin condensation. *Proc. Natl Acad. Sci. USA.* **98**, 10148-10153 (2001).
12. Hanpude, P., Bhattacharya, S., Dey, A.K., and Maiti, T.K. Deubiquitinating enzymes in cellular signaling and disease regulation. *IUBMB Life.* **67**, 544-555 (2015).
13. Fielding, A.B., Concannon, M., Darling, S., Rusilowicz-Jones, E.V., Sacco, J.J., Prior, I.A., Clague, M.J., Urbé, S., and Coulson, J.M. The deubiquitylase USP15 regulates topoisomerase II alpha to maintain genome integrity. *Oncogene.* **37**, 2326-2342 (2018).

Q5. Figure S3: The amino acid sequence is not aligned to the coding sequence. Perhaps the font was converted from a monospace font to a variable width font when generating the figure from a text file?

Response to Q5: We have modified the Figure S3 to align the amino sequence with the coding sequence.

Q6. Figure 3 (b and c): I do not see what these analyses add to the current manuscript? What does a picture of the structure of USP15 or a phylogenetic tree of USP15 protein from different species have to do with the genotype to phenotype association of the synonymous mutation? I do not see

that these contribute anything to this manuscript, and the authors should consider removing them (or provide arguments for why they are necessary in case I missed it in the text). Other figures/tables that I do not find any motive for including are Figure S4, Table S2, Table S6. Or are these used in order to establish that the locus “1881663_28” is really the M. virgate USP15 ortholog? In that case this should be clarified in the text.

Response to Q6: We would remove the protein structure from this manuscript. However, the reason why we hope to keep the phylogenetic analysis of USP15 in our manuscript is to validate the function of *MvUSP15* by demonstrating the domains are highly conserved during evolution. The following are some detailed reasons to support retention of the phylogenetic information.

First, the genome of the mussel has not been officially published, in order to increase the confidence of the sequence, we compared the *MvUSP15* sequence with other species. Second, by comparing with USP15 sequences in diverse organisms from different taxon, we found the domain components in these USP15 sequences are highly conserved, implying that USP15 in different organisms may play important functions, which include the powerful inhibition of mitochondrial autophagy, which we discuss in the current manuscript. Third, based on the results of multiple sequence alignment, a Bayesian phylogenetic tree was constructed to indicate the evolutionary process of USP15 in different taxa (Figure 3c). The *MvUSP15* was clustered together with the USP15 sequences from other Molluscan species; also, species from the clade Lophotrochozoan are clustered together and differentiated with species from the other two major clades of bilaterian animals, namely Arthropoda and Chordata. These results not only indicated that USP15 arose much earlier than previously reported¹⁴, but also illustrate the characteristic conservation in its evolution.

For these reasons, we believe that these figures and tables that contain the list of sequences for phylogenetic tree construction are elements worth retaining in our paper.

References

14. Vlasschaert, C., Xia, X., Coulombe, J., and Gray, D.A. Evolution of the highly networked deubiquitinating enzymes USP4, USP15, and USP11. *BMC Evol. Biol.* **15** (2015).

Q7. While the data show a correlation between the allele at USP15 and USP15 gene expression variability in heat stressed mussels, the mechanism for this is not clear. The authors mention that “Codon usage bias between UCA and UCG contributes to genotypical differentiation in the transcription of MvUSP15”. What is this supposed to mean? How would a single synonymous mutation affect transcription? The discussion further mentions codon usage bias and transcription in the same sentence, without any mention of the most obvious level at which codon usage bias is expected to have an effect: translation. Are the UCA and UCG codons expected to be translated at different efficiencies?

Response to the questions related to transcription in Q7: The effects of codon usage biases (CUB) on gene expression are thought to be mainly due to its impacts on translation including the regulation of translational rate and the contranslational protein folding. However, several promising lines of research point out that CUB could play important roles in gene expression at the transcriptional level. What we hope to summarize using the sentence “Codon usage bias between UCA and UCG contributes to genotypical differentiation in the transcription of *MvUSP15*” is that we wish to demonstrate that the influence of codon usage on mRNA levels is through transcription in a translation-independent manner as Zhao *et al.* (2021) described¹⁵ that codon usage had strong genome-wide correlations with total and nuclear RNA levels. Although their results showed that optimal codons showed positive correlations with genome-wide mRNA levels and rare codons exhibited negative correlations, some research shows that the biological impacts of a single synonymous mutation on mRNA level are pleiotropic. While codon optimization enhances mRNA levels in most cases, the opposite pattern also has been observed. Furthermore, we have investigated the influence of synonymous mutation on mRNA structure by comparing the free energy change occurring during formation of the ensemble of secondary structures (ΔG_{fold}) in two orthologous mRNAs. The absolute ΔG_{fold} value of mRNA using the UCG codon is statistically significantly higher than the other using UCA codon (temperature effects on folding differ as well). This difference is interesting because it has been pointed out that differences in mRNA stability caused by different codon usage have a positive correlation with mRNA expression level¹⁶. We conjecture that the up-regulated *MvUSP15* mRNA levels in some G-type mussels at the temperature of 42 °C at least partly result from the UCG codon usage causing more stable mRNA structure. The particular underlying mechanisms by which synonymous codon usage may influence mRNA level are still under debate. Because our work placed more attention on the ecological importance of adaptive synonymous mutation, the influence of this site on transcription is not the focus of the current research. However, it is another future research direction for us.

We have modified the part of the discussion related to gene transcription as follows:

Page, 10, line 248-257, “We compared the stability of two orthologous mRNAs of *MvUSP15* with the base difference (G or A) at synonymous mutation 1881663_28. The free energy change occurring during formation of the ensemble of secondary structures (ΔG_{fold}) was used as an index for estimating the stability of mRNA structure. We first performed the free energy minimization structure prediction for two orthologous mRNAs from 22 °C (295.15 K) to 52 °C (325.15K) using the Fold program with default parameters in the RNAstructure package. The ΔG_{fold} for two orthologous mRNAs was calculated by using the efn2 algorithm. Calculations were performed with default parameters at different temperatures (22° to 52 °C) along the predicted secondary structure

determined using the Fold program.

Page, 16, line 414-419, “The mRNA with G at the synonymous mutant site had a lower ΔG_{fold} (higher stability) compared with the mRNA with A base at that site at most temperatures. With temperature rising, the effects of the synonymous mutation on ΔG_{fold} differed between the G and A containing orthologs are significant (Paired Mann-Whitney test, $V = 1370.5$, p value < 0.05), indicating the single synonymous mutation having influence on mRNA stability.”

Page, 20-21, line 544-576, “The synonymous mutation examined in this study, wherein the preferred codon UCA was at times replaced by the rare and non-optimal codon UCG⁶⁹, could have led to a number of influences on gene expression. For example, codon usage can have significant effects on translation because synonymous codons vary in translation efficiencies; these effects can influence important aspects of the translation process, for example, amino sequence elongation speed and cotranslational protein folding.

Different synonymous codon usage can also affect transcription levels in a translation-independent manner. Previous study has pointed out that the biological impacts of a single synonymous mutation on mRNA levels are pleiotropic: the usage of optimal codon enhances transcriptional levels at most cases; however, in some instances the usage of non-optimal codon resulted in higher mRNA levels. The latter phenomenon was also found in the present study: although the relatively non-optimal codon UCG occurred in the CDS of *MvUSP15* of G-type mussels, this genotype had stronger up-regulation of transcriptional levels than A-type mussels under conditions of heat stress. Potential explanation for this result might be that the effects of synonymous codon usage on transcription could be complex and involve factors other than codon bias usage. For instance, due to the alteration of synonymous codon usage, the secondary structure and stability of mRNA can be affected, further leading to the transcriptional changes, some of which could be adaptive. In the present study, the synonymous mutation in *MvUSP15* was shown to cause differences in ΔG_{fold} between the A- and G-type mRNAs; the mRNA containing the UCG codon had a higher absolute value of ΔG_{fold} and thus was more stable than the sequence using the UCA codon. According to the studies of Victor *et al.*, who investigating the influence of the codon usage pattern and mRNA structural stability on mRNA expression levels, higher stability of mRNA secondary structure resulting from different codon usage, led to higher mRNA abundance. Thus, the usage of the non-optimal codon UCG, which led to higher mRNA stability may further cause the

up-regulated *MvUSP15* transcriptional level in some G-type mussels experiencing temperature increases. However, a large gap still exists in more specific understanding of the mechanisms explaining how synonymous mutation influences gene expression; a great deal of work combining the approaches of reverse genetics such as an F2 hybrid screen with quantitative trait locus (QTL) mapping should be useful for further investigating the role of synonymous mutations in the thermal adaptation.”

References

15. Zhao, F. et al. Genome-wide role of codon usage on transcription and identification of potential regulators. *Proc. Natl Acad. Sci. USA*. **118**,6 (2021).
16. Victor, M. P., Acharya, D., Begum, T. & Ghosh, T. C. The optimization of mRNA expression level by its intrinsic properties-Insights from codon usage pattern and structural stability of mRNA. *Genomics*. **111**, 1292-1297 (2019).

Q8. The manuscript could need some improvement of the English language. Most sentences are ok, but here are some examples of sentences that needs to be improved. Note that I have not looked very carefully and have probably missed several other similar language problems:

Response: The writing has been modified in the revised manuscript by a native English speaker.

Line 389 - “One of the convincing cellular physiological functions of USP15 has been shown as a powerful inhibitor against PINK1/Parkin-mediated mitochondrial apoptosis pathway via ubiquitination antagonizing” (I guess it is meant to say “by antagonizing ubiquitination”?)

Line 413 – “Another study focusing on synonymous mutations of IGF-1 in mammals suggested that the occasional using rare codons inducing slightly upregulation in the transcriptional level might due to the usage of a rare codon relieved the pressure of commonly used codons to a certain extent” (there are several problems with the structure and grammar of the sentence that makes it difficult to figure out exactly what it is meant to say).

Page, 21, line 551-554, “A previous study has pointed out that the biological impacts of a single synonymous mutation on mRNA levels are pleiotropic: the usage of optimal codon enhances transcriptional levels at most cases; however, in some instances the usage of non-optimal codon resulted in higher mRNA levels”

Line 407 – “Gerdol et al. (2015) found species belonging to Mytilidae had codon usage bias towards UCA codon rather than UCG codon and our characterization analysis of the MvUSP15 cDNA sequence was consistent with this opinion” (the word “opinion” makes it look like there is a matter of debate with regards to the codon bias that Gerdol et al reported. I guess it is just a poor choice of words, and that “observation” would be a better choice?)

Page, 20, line 544-545, “The synonymous mutation examined in this study, wherein the preferred codon UCA was at times replaced by the rare and non-optimal codon UCG”

Reviewer #2 (Remarks to the Author):

MAJOR COMMENTS:

I greatly appreciate the efforts and work of the authors and do believe they have very interesting findings toward a genetic basis of heat tolerance in this Mytilus species. However, I find two logical flaws in their interpretation of the results. Without these flaws addressed by additional experimental work, my suggestion is the rejection of the manuscript in its current form.

Response: Thanks for your comments and appreciation of our work. We have modified our manuscript based on your suggestions as follows.

1. First, the authors show no evidence that the synonymous polymorphism in MvUSP15 are the mechanistic basis of the of the differential thermal tolerance observed. I find it at least as probably that the genetic basis of the phenotype is another site that is in tight linkage with the identified locus.

Response to Q1: We appreciate your criticism and realize that the lack of detailed description of methods may underlie some of the problem you mention. In the present work, we use Arrhenius breakpoint temperature (ABT) of mussels as an indicator for determining when sublethal physiological stress occurs. In general, heart rate increases with temperature increase until the ABT is reached, after which heart rate decreases rapidly and anaerobic metabolism is activated. Our results of genome-wide phenotype-genotype association analysis has shown that the synonymous polymorphism in MvUSP15 has a significant correlation with the ABT temperature of mussels. Therefore, we believe that this synonymous mutant site is potentially one genetic basis of thermal tolerance in mussels.

We also share your belief that this synonymous polymorphism in MvUSP15 is not the only mechanistic basis of the differential thermal tolerances in these mussels. However, in the dataset for this work, we have performed multiple analyses to demonstrate that this polymorphism has the maximum possibility to influence the thermal tolerance compared with other sites. Our evidence concerning this point is as follows: (i), this site has the highest p-value corrected by double correction methods among other sites. And it is located on the seventh exon of the Chr12.1841 gene, so the mutation of this site is very likely to directly affect the expression of the gene. Besides, Chr12.1841 has been annotated with a relatively clear function, namely ubiquitin carboxyl-terminal hydrolase 15, which is involved in pathways such as mitophagy which is considered to be associated with heat tolerance. (ii), we have also analyzed other sites with significant p-values (p-value < 1e⁻

⁴), including 2020699_47 at pseudo-chromosome 13, 1796187_33 at pseudo-chromosome 12, 1652060_107 at pseudo-chromosome 11; however, these sites are located at introns of genes and more importantly, the annotation results of the genes at which these three sites are located were not clear, so we cannot get exact information about how these genes might help to regulate the heat tolerance of mussels. (iii), we also have conducted linkage disequilibrium-based SNP pruning analysis in this work using PLINK software (version 1.90). The loci under linkage disequilibrium (r -squared > 0.2) within 50kb range have been pruned, and the potential linkage disequilibrium-causing influence by adjacent sites was eliminated. And the long-distance linkage disequilibrium between chromosomes was not taken into consideration in this work for the reason that the mussel genome we used is a draft genome, which contains 14 pseudo-chromosomes of which the numbers are arranged in ascending order of chromosome size, and thus do not have true biological meanings. Therefore, we put our focus on the investigation of how the synonymous locus 1881663_28 at MvUSP15 can affect heat tolerance of the mussel.

To address these issues, we have modified our manuscript as follows.

Page 7, line 161-169, “The Arrhenius Breakpoint temperature (ABT) for cardiac performance, the temperature at which the HR decreases sharply with progressive heating and after which heart rate decreases rapidly, was determined using a regression analysis method that generates the best fit line on either side of a putative break point for the relationship of ln-transformed heart rate (beats per minute) against absolute temperature and calculated using segmented package in R version 4.1.0. ABT temperature was an indicator for determining when anaerobic metabolism is activated by sublethal physiological stress, and has been used as a common index for evaluating sublethal thermal tolerance in molluscs”

Page 14, line 364-373, “Apart from the locus 1881663_28, three other sites showed significant correlation with heat tolerance of mussels; these were, the locus 2020699_47 at pseudo-chromosome 13 (genomic-control corrected p-value = $1.55e^{-5}$), 1652060_107 at pseudo-chromosome 11 (genomic-control corrected p-value = $7.699e^{-5}$), and 1796187_33 at pseudo-chromosome 12 (genomic-control corrected p-value = $7.278e^{-5}$). However, these loci were located in the introns of genes whose functions could not be determined based on the genome annotation files and BLAST results. No strong linkage disequilibrium relationship was found between the synonymous mutant locus 1881663_28 and locus 1796187_33 ($D' = 0.2772$, $r^2 = 0.033$).”

Q2. Second, the authors posit that codon usage impacts MvUSP15 expression and examine expression at the transcriptional level. Codon bias typically affects translation, but I did follow up on the author's discussion of Wang et al. (2019; <https://doi.org/10.1016/j.omtn.2021.08.007>), in which mRNA stability appears to be impacted by codon usage.

Additional experiments to map the mechanistic locus underlying the phenotype could include an F2 hybrid screen with QTL mapping using a WGS approach to identify candidate loci at higher resolution than the current RAD-based mapping, ideally followed up by reverse genetics to test the candidates.

*It appears that forward genetics are not yet published for *Mytilus* species, but in the absence of genetic manipulation, there are additional experiments which could be performed, perhaps examining mRNA stability by blocking transcription using Actinomycin-D and assessing mRNA levels.*

Response to Q2: Thank you very much for suggesting the F2 hybrid screen and QTL mapping. As the mussel *Mytilisepta virgata* is a non-model organism and has not been commercially farmed so far, it is impossible for us to complete a whole generation indoor cultivation for now.

We have closely considered your suggestions on the additional experiments for investigating the relationships between this synonymous mutant locus and the stability of MvUSP15 mRNA. There are not at present procedures for developing cell lines from mussels for cell culture, and in living mussels it would be hard to inject a specific amount of Actinomycin-D to regulate translational activity. Thus, examining mRNA stability based on transcriptional inhibition methods is hard to perform in this species. However, according to published studies¹⁷, the free energy change (ΔG_{fold}) that occurs during formation of RNA secondary structure can be used as an index of intrinsic stabilities, and synonymous mutation can adaptively modify the stability of mRNAs without altering stability or function of the proteins they encode. Thus, the transformation of G to A might change mRNA stability of *MvUSP15*. We have utilized the methods for determining mRNA stability used in the research cited above and compared the stabilities of the two orthologous mRNA of *MvUSP15* with the difference in usage of UCG or UCA codon by predicting free energy minimization structure using the Fold program and calculating the ΔG_{fold} using the efn2 algorithm in the RNAstructure package¹⁸. We found that the synonymous mutation did produce significant differences between the ΔG_{fold} of these two orthologous mRNA, indicating that the changes at this site might have significant effects on *MvUSP15* mRNA stability. The differences in stability of mRNA secondary structure affected by different codon usage could have a positive correlation with mRNA levels¹⁹. Thus, we conjecture that the usage of non-optimal codon UCG, which leads to higher mRNA stability may further cause an up-regulated MvUSP15 transcriptional level in some G-type mussels. We have modified our manuscript as follows:

Page, 10, line 248-257, “We compared the stability of two orthologous mRNAs of MvUSP15 with the base difference (G or A) at synonymous mutation 1881663_28. The free energy change occurring during formation of the ensemble of secondary structures (ΔG_{fold}) was used as an index for estimating the stability of mRNA structure. We first performed the free energy minimization

structure prediction for two orthologous mRNAs from 22 °C (295.15 K) to 52 °C (325.15K) using the Fold program with default parameters in the RNAstructure package. The ΔG_{fold} for two orthologous mRNAs was calculated by using the efn2 algorithm. Calculations were performed with default parameters at different temperatures (22° to 52 °C) along the predicted secondary structure determined using the Fold program.

Page, 16, line 414-419, “The mRNA with G at the synonymous mutant site had a lower ΔG_{fold} (higher stability) compared with the mRNA with A base at that site at most temperatures. With temperature rising, the effects of the synonymous mutation on ΔG_{fold} differed between the G and A containing orthologs are significant (Paired Mann-Whitney test, $V = 1370.5$, p value < 0.05), indicating the single synonymous mutation having influence on mRNA stability.”

Page, 20-21, line 544-576, “The synonymous mutation examined in this study, wherein the preferred codon UCA was at times replaced by the rare and non-optimal codon UCG⁶⁹, could have led to a number of influences on gene expression. For example, codon usage can have significant effects on translation because synonymous codons vary in translation efficiencies; these effects can influence important aspects of the translation process, for example, amino sequence elongation speed and cotranslational protein folding.

Different synonymous codon usage can also affect transcription levels in a translation-independent manner. Previous study has pointed out that the biological impacts of a single synonymous mutation on mRNA levels are pleiotropic: the usage of optimal codon enhances transcriptional levels at most cases; however, in some instances the usage of non-optimal codon resulted in higher mRNA levels. The latter phenomenon was also found in the present study: although the relatively non-optimal codon UCG occurred in the CDS of *MvUSP15* of G-type mussels, this genotype had stronger up-regulation of transcriptional levels than A-type mussels under conditions of heat stress. Potential explanation for this result might be that the effects of synonymous codon usage on transcription could be complex and involve factors other than codon bias usage. For instance, due to the alteration of synonymous codon usage, the secondary structure and stability of mRNA can be affected, further leading to the transcriptional changes, some of which could be adaptive. In the present study, the synonymous mutation in *MvUSP15* was shown to cause differences in ΔG_{fold} between the A- and G-type mRNAs; the mRNA containing the UCG codon had a higher absolute value of ΔG_{fold} and thus was more stable than the sequence using the UCA

codon. According to the studies of Victor *et al.*, who investigating the influence of the codon usage pattern and mRNA structural stability on mRNA expression levels, higher stability of mRNA secondary structure resulting from different codon usage, led to higher mRNA abundance. Thus, the usage of the non-optimal codon UCG, which led to higher mRNA stability may further cause the up-regulated *MvUSP15* transcriptional level in some G-type mussels experiencing temperature increases. However, a large gap still exists in more specific understanding of the mechanisms explaining how synonymous mutation influences gene expression; a great deal of work combining the approaches of reverse genetics such as an F2 hybrid screen with quantitative trait locus (QTL) mapping should be useful for further investigating the role of synonymous mutations in the thermal adaptation.”

References

17. Liao, M., Dong, Y. & Somero, G. N. Thermal adaptation of mRNA secondary structure: stability versus lability. *Proc. Natl Acad. Sci. USA*. **118**, e2113324118 (2021).
18. Mathews, D. H., Sabina, J., Zuker, M. & Turner, D. H. Expanded sequence dependence of thermodynamic parameters improves prediction of RNA secondary structure. *J. Mol. Biol.* **288**, 911-940 (1999).
19. Victor, M. P., Acharya, D., Begum, T. & Ghosh, T. C. The optimization of mRNA expression level by its intrinsic properties—Insights from codon usage pattern and structural stability of mRNA. *Genomics*. **111**, 1292-1297 (2019).

MINOR COMMENTS:

(1) How was the *MvUSP* identified from the candidate loci? Are there additional candidate loci?

Response to (1): We first apologize for our negligence to provide adequate description of certain of our methods. Apart from the draft genome of *Mytilisepta virgata*, we also got the annotation files comprising the description of gene function. Thus, after we perform gene-based annotation using ANNOVAR and obtain the information on which gene the synonymous mutation occurs, we can refer to the annotation files to gain some insight about the potential gene function. Besides, the nucleotide sequence of CDS and amino sequence of protein are also used to blast against the dataset of NCBI and Uniprot website to further confirm the gene function. Also, the molecular phylogeny relationship of the gene with other homologous sequences, as well as the inspection of conserved domains obtained by searching on the SMART website was performed. Finally, we find the synonymous mutant locus locates on the gene with the function of ubiquitin carboxyl-terminal hydrolase 15, which is named as *MvUSP15* in this research.

As for whether there are other sites within *MvUSP15* that could be of interest, we did find other four base site changes within the region of *MvUSP15*, i.e. 1881663_64, 1881700_56, 1881700_132, 1881700_150. However, the loci 1881663_64, 1881700_56, 1881700_132 were pruned out during the LD-based SNP data filtering process. The remained locus 1881700_150 after LD-based SNP data filtering shows no significant correlation with heat tolerance (genomic-control corrected p-value = 0.4578), and the same situation occurs in its linked site: 1881700_56, 1881700_132. Among these three mutant sites, the loci 1881663_64 is the site most likely to be another site of interest for its possible linkage relationship with 1881663_28. However, we also used the SNP data before LD-based SNP data filtering (only using a minimum number of populations of three, a maximum number of missing samples per locus of 20%, and a minor allele frequency of 0.05, a maximum per-SNP missing percentage of 10% and exact Hardy-Weinberg disequilibrium p-value less than 1e-6) to perform the association analysis again and found its sites having no significant association with heat tolerance (genomic-control corrected p-value = 0.006328). Therefore, these mutant sites are not listed as candidate loci for affecting the heat tolerance of the mussel.

Page, 8, line 199-203, “The function of the gene at which the mutant sites were located was firstly confirmed by referring to the annotation file of the *M. virgata* genome, and then the exact function of the gene was further classified based on BLAST results, molecular phylogeny and inspection of conserved domains.”

Page, 14, line 351-357, “We also found that there were four mutant sites within the genomic region of *MvUSP15* gene; these were, the locus 1881663_64 in the seventh exon of *MvUSP15*, the loci 1881700_56, 1881700_132 and 1881700_150 in the intron of *MvUSP15*. However, the loci 1881663_64, 1881700_56, 1881700_132 had been pruned out by SNP filtering and thus not included in association analysis. The remained site 1881700_150 showed no significant correlation with heat tolerance (genomic-control corrected p-value = 0.4578).”

(2) How large is the window based upon knowledge of linkage in this/related taxa?

Response to (2): The parameters used in LD-based SNP pruning were from the protocol “Data quality control in genetic case-control association studies”²⁰, in which the windows size is set as 50 Kb. This size is also used in many other works involving organisms across taxa²¹⁻²⁴.

References

20. Anderson, C. A. et al. Data quality control in genetic case-control association studies. *Nat. Protoc.* **5**, 1564-1573 (2010).
21. Nakamura, Y. et al. A genome-wide association study on adherence to low-carbohydrate diets

- in Japanese. *Eur. J. Clin. Nutr.* **76**, 1103-1110 (2022).
22. Caragiulo, A. et al. Coyotes in New York City Carry Variable Genomic Dog Ancestry and Influence Their Interactions with Humans. *Genes*. **13**, 1661 (2022).
 23. Derbyshire, M. C. et al. A whole genome scan of SNP data suggests a lack of abundant hard selective sweeps in the genome of the broad host range plant pathogenic fungus *Sclerotinia sclerotiorum*. *PLoS One*. **14**, e214201 (2019).
 24. Crombie, T. A. et al. Local adaptation and spatiotemporal patterns of genetic diversity revealed by repeated sampling of *Caenorhabditis elegans* across the Hawaiian Islands. *Mol. Ecol.* **31**, 2327-2347 (2022).

(3) Regarding the qPCR experiments, do the experiments meet MIQE standards?

Response: The qPCR experiments strictly obey the MIQE standards ²⁵.

First, experimental design: the experimental group contained the mussels experiencing heat treatment, which is at the rate of 6°C h⁻¹ from 22 °C to 42 °C, while the control group contained the mussels without experiencing heating and were kept at 22°C in air during the heating period of the experimental group. The numbers of mussels in experimental and control groups are 135 and 36, respectively.

Second, sample collection: all the mussels used for experiments were sampled randomly from their natural habitats on July 31st, 2021 and transported back to the laboratory within 3h for a two-month acclimation period. After the heating experiments, all the mussels were immediately frozen using liquid nitrogen. The frozen mussels were dissected to remove their adductor muscle. The tissues were then stored at -80 °C and used within two days for the following procedures.

Third, total RNA extraction: the total RNA was extracted from the adductor muscle of mussels in both the heated-treated group and the control group using TRIzol reagent (Invitrogen, China) following the manufacturer's protocol. More precisely, all steps of extraction were performed at room temperature (20–25°C) and RNaseZap (RNase Decontamination Solution) was used to remove RNase contamination from work surfaces. The RNA purity was determined by measuring the A260/A280 ratio with a Nanodrop spectrophotometer (Thermo Scientific, China). RNA quality was assessed by 1% agarose gel electrophoresis.

Fourth, reverse transcription: the high-quality isolated RNA was used immediately for cDNA synthesis. The reverse transcription was conducted using PrimeScript RT reagent Kit with the gDNA Eraser (Takara, China) following the manufacturer's protocol. More precisely, the genomic DNA elimination reaction was conducted first. The reaction solution was prepared on ice. A master mix was prepared for components other than the RNA sample in a volume sufficient for the number of reactions plus 2. Then the mix was dispensed at an appropriate volume (total volume minus total RNA volume) into a microtube before the RNA sample was added. The procedures above were

conducted on ice. Each microtube contained about 1 µg of total RNA, 1.0 µl gDNA Eraser, 2.0 µl 5X gDNA Eraser Buffer and a volume of Rnase-free water to bring volume up to 10 µl. Then the reaction was conducted at the temperature of 42°C for a period of 2 minutes. After the genome DNA was removed, the reverse-transcription reaction was performed. A master mix containing all components except the genomic DNA elimination reaction solution was prepared in a sufficient volume for the number of reactions plus 2. Then 10 µl of master mix was added to the microtubes containing the reaction solution from previous step and the solution was then mixed gently. The procedures above were conducted on ice. Each microtube contain about 10 µl reaction solution from the previous step, 4.0 µl 5X PrimeScript Buffer 2 (for Real Time), 1.0 µl PrimeScript RT Enzyme Mix I, 1.0 µl RT Primer Mix, and 4.0 µl Rnase-free water to make the volume up to 20 µl. Then the reaction was performed at 37°C for 15 minutes, 85°C for 5 seconds and immediately used for Real-Time PCR.

Fifth, qPCR procedure: specific primers (Qpcr1) for MvUSP15 were first designed based on the identified sequence and the elongation factor 1 α (EF-1 α) was chosen as the reference gene for internal standardization according to previous research²⁶. These two pairs of primer sequences were validated by melting curve analysis to ensure there were no nonspecific product amplification and primer-dimer formation. The PCR efficiencies for EF1 α and MvUSP15 were calculated by standard curve analysis and were 108.2% and 105.2%, respectively. 2 \times SYBR Premix Ex Taq was used for qPCR procedure following the manufacturer's protocol (TaKaRa, China). More precisely, the qPCR reaction system contained 5.0 µl TB Green Premix Ex Taq (Tli RNaseH Plus) 2X, 0.4 µl PCR Forward and Reverse Primer, 2.0 µl product of RT-PCR which contained cDNA templates and had been diluted 10 times, and 2.2 µl RNase-free water. The qPCR reaction was performed on CFX96 Real Time PCR Detection System (Bio Rad) following this procedure: 95°C for 30 s, followed by 40 cycles of 95°C for 5 s and 60°C for 30 s, then 95°C for 10s and the melt curve analysis was then conducted with 0.5°C increments from 65 to 95°C for 5 s per procedure. No template control (NTC) was also performed to prevent any DNA or RNA template from a reaction serving as a control for extraneous nucleic acid contamination, and to serve as an important control for primer dimer formation. Each sample was tested in triplicate.

Finally, data analysis: the relative expression level of MvUSP15 was normalized and calculated using the $2^{-\Delta\Delta Ct}$ method²⁷ performed on Bio-Rad CFX Manager.

References

25. Bustin, S. A. et al. The MIQE Guidelines: Minimum Information for Publication of Quantitative Real-Time PCR Experiments. *Clin. Chem.* **55**, 611-622 (2009).
26. Gerdol, M. et al. The purplish bifurcate mussel *Mytilisepta virgata* gene expression atlas reveals a remarkable tissue functional specialization. *BMC Genoms.* **18**, (2017).
27. Livak, K. J. & Schmittgen, T. D. Analysis of relative gene expression data using real-time quantitative PCR and the 2(-Delta Delta C(T)) Method. *Methods.* **25**, 402-408 (2001).

(4) Is *EF-1 α* stable and suitable for use as a housekeeping/internal control gene based on empirical data from this species?

Response: Elongation factor 1 α (EF-1 α) was chosen as the reference gene for this study for the reason that it has been described previously as a housekeeping gene without displaying high tissue specificity⁶. Primer sequences were also validated in this study by melting curve analysis to ensure there was no nonspecific product amplification or primer-dimer formation. The following are the melt curve and melt peak during amplifying EF-1 α .

The explanation of *T₉₉* and ADM would improve the clarity of the manuscript.

Response: We have modified the text by adding the explanation of *T₉₉* and ADM. The precise modification is as follows:

Page, 13, line 330-332, “The *T₉₉* was the monthly highest temperature that the mussels are likely to experience and was used as a measure of ‘acute’ thermal stress in each month, while ADM was considered as a measure of ‘chronic’ high-temperature exposure”.

We also have added the description on the process of heart rate measurement and the ddRAD library construction to improve the comprehensibility of the manuscript. This added content is in the Methods and Materials section as follows:

Page, 6-7, line 145-179, “From February to December 2016, *M. virgata* specimens were randomly collected every 2 months on rocky shore in Dongshan Island, Fujian Province (117°29’ E, 23°39’N). After collection, mussels were transported back to the laboratory within 3h, placed in a

plastic basket, and immersed in 20 l fresh seawater with temperature of $\sim 20^{\circ}\text{C}$ and salinity of ~ 33 psu to simulate the in situ annual average water temperature for a 3-day acclimation period before performing the heart rate measurements. Seawater was aerated continuously and exchanged daily. Mussels were fed daily with concentrated *Chlorella*. For the heart rate measurement experiments, which used a non-invasive method²⁰, the designated temperatures increased at a rate of $6^{\circ}\text{C}\cdot\text{h}^{-1}$ in air from acclimation temperature ($\sim 20^{\circ}\text{C}$) until the heart rate fell to zero. For detecting the heartbeat of mussels, an infrared sensor fixed (with Super Glue, Loctite, USA) to the mussel shell at a position above the heart (next to the mid-dorsal posterior hinge area) was used. Then the variations in the light-dependent current produced by the heartbeat were amplified, filtered and recorded using an infrared signal amplifier (AMP03, Newshift, Portugal) and PowerLab AD converter (8/30, ADInstruments, Australia), and finally viewed and analyzed using LabChart v7 (ADInstruments, Australia). The Arrhenius Breakpoint temperature (ABT) for cardiac performance, the temperature at which the HR decreases sharply with progressive heating and after which heart rate decreases rapidly, was determined using a regression analysis method that generates the best fit line on either side of a putative break point for the relationship of ln-transformed heart rate (beats per minute) against absolute temperature and calculated using segmented package in R version 4.1.0^{30,31}. ABT temperature was an indicator for determining when anaerobic metabolism is activated by sublethal physiological stress, and has been used as a common index for evaluating sublethal thermal tolerance in molluscs²⁴⁻²⁶.

For identifying single nucleotide polymorphisms (SNPs) in a total of 64 extracted DNA samples (21 collected in April, 23 collected in August, and 20 collected in December), double digest restriction site-associated DNA sequencing (ddRADseq) was performed. Sequencing libraries were prepared using the EcoRI and MspI restriction enzymes (New England Biolabs, MA, USA) and a protocol adapted from the McDaniel Laboratory at the University of Florida (<https://mcdaniellab.biology.ufl.edu/data/>). Libraries were sequenced on an Illumina NovaSeqs 150 bp paired-end reads at Berry Genomics Corporation (Beijing, China), and deposited in BioProject (PRJNA517974) and BioSample (SAMN10849586 - SAMN10849649) databases at NCBI (National Center for Biotechnology Information).”

REVIEWERS' COMMENTS:

Reviewer #1 (Remarks to the Author):

I think my questions and concerns have been (mostly) properly addressed, and that the revised manuscript is significantly improved by the changes.

There are, however a couple of questions that were probably misunderstood (see the original report for the actual questions):

Q4: I think the authors misunderstood my line of questioning regarding the possibility for additional sites that could affect mRNA levels. I wrongfully used the term "chromatin condensation" when I probably should have used "chromatin remodeling", which is probably why my question was misunderstood. What I was after was if the site of the synonymous mutation could possibly affect transcription, not due to being in the coding sequence but by being in some other sequence element that just happen to overlap with the coding sequence?

- Anyway, this was not a huge issue but if you are given another chance to revise (or if you submit somewhere else) you could consider this.

Q7: Here I think I was the one that misunderstood. When you wrote "transcriptional" I thought you meant a single snp in a several kb long transcript had an effect on the amount of full-length mRNA *produced*, when you only meant the snp had an effect on the level of mRNA *detected* (which includes both transcription and post-transcriptional processes, including differential decay of slowly translated vs. quickly translated mRNAs).

That a mutation can have an effect on mRNA stability is not as far-fetched, but then you should call it "RNA stability" and not "transcription". To me "transcription" is only the process of transcribing DNA into RNA. In my mind, anything that happens to the RNA after transcription is "post-transcriptional". Besides this, I doubt RNA stabilities estimated by *in silico* folding are useful for saying anything about the *in vivo* stability of an mRNA (at least not an actively translated mRNA that would be protected by ribosomes). Is there always a straight-forward link between RNA folding stability and RNA decay?

If you want to argue the effect of the synonymous mutation is on some other level than translation you need to demonstrate that translation is not needed to see the effect. I understand blocking translation or transcription with drugs could be tricky to do in whole organisms, so I'm not going to suggest that. But would it not be possible to do nuclear RNA extracts and at least being able to determine in which compartment (nucleus or cytoplasm) the effect is seen? If the effect is indirect trough an effect on translation one would expect no difference in mRNA amount in the nucleus, while if the effect is on transcription a similar difference should be seen in both compartments (I just added this idea for your consideration and do not ask of you to do this at this point).

- I suggest toning down the RNA stability argument (as you have not done any experiments to test this hypothesis) and that you refrain from using "transcriptional" to mean anything other than the process of transcription. Or just make it clear that you mean the effect is on the level of RNA concentration, which could be anywhere between transcription initiation and mRNA decay (with or without the involvement of translation).

- I also suggest that you make it clear that your qPCRs were not made in such a way that you could determine at which level (transcription or decay, with or without the involvement of translation) the RNA levels were affected.